# Molecular QTL are enriched for structural variants in a cattle long-read cohort
Xena Marie Mapel[1,2], Alexander S. Leonard [1,2] & Hubert Pausch [1] ✉

Sequencing cohorts with long-read technology is crucial to understand the impact of structural variants (SVs) on complex traits. Here, we obtain 4.86 terabases of HiFi reads with an average read N50 of 16.3 Kb from 120 *Bos taurus taurus* bulls, yielding a mean coverage depth of 13.5-fold. We genotype 23.8 M small variants (SNPs and short INDELs) and 79.3 k SVs to perform association testing with molecular phenotypes derived from a subset of 117 bulls with total RNA sequencing data from testis tissue. We identify 27.3 k molecular QTL (molQTL) including 316 for which SVs were the most significant variant. This corresponds to a 2.1- and 5.6-fold enrichment of SVs among expression and splicing QTL, respectively. When considering SVs in perfect LD with the lead small variant, the enrichment increases to 6.1- and 12-fold for expression and splicing QTL in testis, respectively. Imperfect genotyping for SVs limits our ability to detect all SV molQTL, suggesting that the true enrichment of SVs among molQTL may be even higher. These results demonstrate that SVs have a profound impact on gene expression and splicing variation but highlight the necessity of improved SV genotyping to fully leverage long-read sequencing cohorts for dissecting complex traits.

Understanding the genetic basis of complex traits and diseases requires comprehensive exploration of genomic variation within and between individuals[1,2]. Genome-wide association studies (GWAS) have identified numerous loci linked to specific traits but require accurate genotypes to detect putative causal variants and estimate allele substitution effects[3]. Single-nucleotide polymorphisms (SNPs) derived from microarrays or short-read sequencing serve as the predominant source of genomic data for GWAS[4–6]. Structural variants (SVs), specifically, insertions, deletions, inversions, and duplications larger than 50 bp, contribute more to genomic variation between individuals than SNPs and other small variants[7]. Short-read sequence data has inherent limitations in detecting and genotyping SVs and thus, until recently, a substantial portion of genomic variation remained undercharacterized[8–10].

Long-read sequencing enables the analysis of DNA fragments orders of magnitude longer than short reads, improving SNP calling in challenging regions[11] and offering the resolution necessary to identify and genotype SVs[12–14]. Studies in humans and plants have begun to explore SV diversity at the cohort level using long-read sequencing. While these efforts remain limited in number, they have provided compelling evidence that SVs disproportionately contribute to phenotypic variation[15–17]. These studies also demonstrated that accurately genotyping large and complex SVs remains challenging, even with long and highly accurate reads[18,19]. However, benchmarking results from human studies may not generalise to other species, since most of the widely applied variant genotyping algorithms are trained on human data and may perform suboptimally elsewhere[11,20,21].

Research on SVs in cattle has primarily relied on pangenome analyses[22–24], short-read sequencing[25–27], and quantitative trait loci (QTL) fine-mapping[28,29]. These studies have revealed several SVs with large impacts on phenotypic traits, including polledness[30], coat colour variation[29,31–33], milk production[34], mastitis susceptibility[28], and fertility[35]. The discovery of such impactful SVs highlights their biological and economic relevance underscoring the critical need to systematically investigate them in genome-wide analyses. Association testing with molecular phenotypes and SVs called from moderate-coverage short-read sequence data (average of 13.8-fold) has identified hundreds of SVs that impact gene expression and splicing in cattle testis tissue[23,25]. However, short-read datasets are limited in their ability to accurately identify and genotype SVs, particularly large or multiallelic insertions and deletions[25–27]. Cohort-scale long-read sequencing in cattle has yet to be conducted to reveal the true impact of SVs on gene expression and splicing.

Here, we assess genetic variation in a cohort of 120 bulls from moderately covered HiFi reads. We identify and genotype 23.8 M small variants and 79.3 k SVs larger than 50 bp representing the vast majority of genetic variants segregating in that population. We perform molecular QTL (molQTL) mapping with deeply sequenced total RNA from testis tissue of the same individuals, identifying more than 27 k gene expression and

---

[1]Animal Genomics, ETH Zurich, Zurich, Switzerland. [2]These authors contributed equally: Xena Marie Mapel, Alexander S. Leonard.
✉e-mail: hubert.pausch@usys.ethz.ch

splicing QTL. Our findings show that molQTL are enriched for SVs but also highlight that obtaining accurate genotypes for SVs remains challenging.

## Results

### HiFi sequencing, alignment, and variant calling of 120 cattle generates an exhaustive set of variants

We resequenced a biobanked cohort of 120 *Bos taurus taurus* samples of primarily Braunvieh (BV) ancestry with existing short read sequencing data[36] with long reads to assess both small variant and SV diversity. Short-read data from the same cohort were previously utilised to genotype small variants and SVs for association analyses with molecular phenotypes[25,36,37]. We collected 4.86 terabases of HiFi reads sequenced from 49 and 41 Sequel IIe 8 M and Revio 25 M SMRT cells, respectively. The mean depth of coverage was 13.8-fold and 13.5-fold for the Illumina and HiFi reads, respectively (Fig. 1a), enabling a fair comparison for variant discovery, although coverage differed substantially between both sequencing technologies for some individuals. Read N50 (mean: 16.3 Kb) and Phred quality score (mean: 33.7) were strongly negatively correlated (Pearson's $r$: −0.80, $p = 5.87 \times 10^{-37}$; Supplementary Fig. 1), as expected for HiFi reads.

We aligned the 120 HiFi and Illumina samples to the ARS-UCD2.0 *Bos taurus* reference genome, which included a recent telomere-to-telomere assembly of a Y chromosome from a Wagyu bull, whereas the remaining genome is from a Hereford cow[38]. Alignment depth of both the short and long-read sequencing was comparable across the autosomes, with only minor increase in the fraction of autosomal sequence covered by alignments suitable for variant calling (MAPQ > 5 and read depth >2) for HiFi reads (99.4% versus 99.0%; Fig. 1b). Conversely, the alignment improvement was noticeable for the X chromosome (90.0% versus 86.0%) and substantial for the Y chromosome (58.9% versus 32.0%). In the former case, this was largely due to the improved mapping of HiFi reads distinguishing high sequence similarity of X:19309231-28499819 to the unplaced contig NKLS02002208.1, while the latter was due to a large fraction of sequence containing higher order repeats, which was almost completely inaccessible with short-read sequencing (Supplementary Fig. 2a, b). However, the male-specific region of the Y chromosome (MSY) still displayed uneven and inconsistent alignment with the HiFi reads due to the repeat structure typically exceeding the length of the sequenced long reads which poses substantial challenges to unambiguous read alignment.

We called small variants with DeepVariant separately for HiFi and Illumina reads, again finding minor differences in the number of detected variants and called genotypes on autosomes and larger differences on sex chromosomes (Table 1). A substantially higher proportion of singletons among the Illumina-only variants may indicate an elevated variant discovery error rate in short-read alignments (Supplementary Fig. 2c, d). The increased number of variants called on X from the HiFi alignments was evenly distributed across the chromosome, while the larger number of variants discovered on Y primarily resulted from the increase in bases covered with HiFi reads in the MSY, with a substantial increase in variants called in previously inaccessible regions that were almost entirely uncalled from short read sequencing. Given all the samples are male, we specified that variants could have heterozygous genotypes in the pseudo-autosomal region (PAR) of the Y chromosome, while they could only be hemizygous in the X or MSY regions. However, there were 'real' signals of heterozygous variants in the hemizygous region, primarily due to copy number variants of amplicon genes (e.g. *TSPY*, *HSFY2*, and *RBMY*) relative to the reference, where the variant allele frequency could estimate the copy number in addition to coverage-based estimates (Supplementary Fig. 3).

We hypothesised that the improved mappability of HiFi reads and the larger number of called variants provide access to previously unidentified genetic variants that may alter gene products or are deleterious to protein function. We assessed potential functional consequences of small variants with the Ensembl Variant Effect Predictor (Table 1), finding 247 and 166 biallelic SNPs with 'HIGH' impacts respectively, private to HiFi- and Illumina-based variant calls, with 2013 common to both. After manual

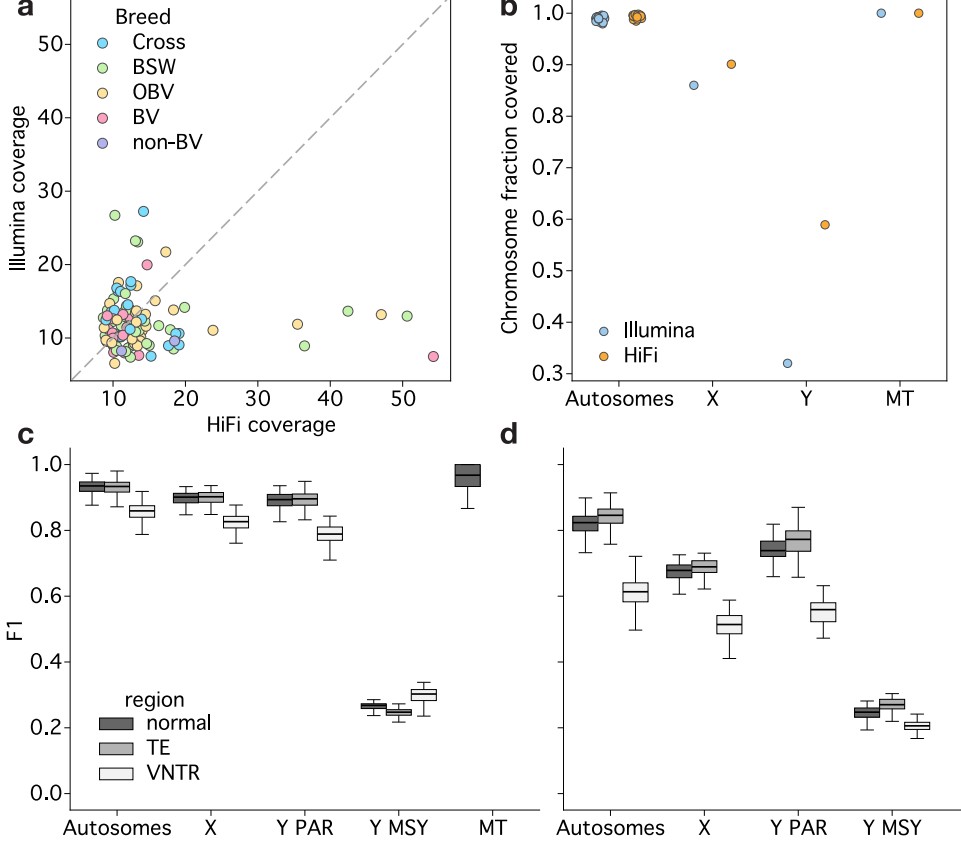

**Fig. 1 | Comparison of alignment coverage and variant accuracy between Illumina and HiFi reads in 120 samples. a** HiFi and Illumina sequencing depth was roughly comparable across bulls with breeds assigned as Original Braunvieh (OBV), Brown Swiss (BSW), ambiguous Braunvieh ancestry (BV), crosses between Braunvieh and a non-Braunvieh breed (Cross), or animals without Braunvieh ancestry (non-BV). **b** Fraction of chromosomes covered by at least two reads with MAPQ ≥ 5 to enable variant calling. F1 score for SNPs (**c**) and indels (**d**), taking the HiFi variants as truth. Variants are stratified by regions annotated as tandem repeats (VNTR), transposable elements (TE; SINE/LINE/LTR), or neither (normal). The Y chromosome is separated by the pseudo-autosomal region (Y PAR) and the male-specific Y (Y MSY).

**Table 1 | Variants called by DeepVariant from the HiFi or Illumina alignments**

|  | Autosomes | X | Y PAR | MSY | MT |
|---|---|---|---|---|---|
| Illumina | 22,636,961 (2,198) | 383,835 (20) | 102,570 (4) | 20,177 (0) | 379 (8) |
| HiFi | 23,163,376 (2,273) | 444,306 (24) | 112,420 (6) | 69,402 (1) | 386 (8) |
| Increase | 2.3% (3.4%) | 15.8% (20.0%) | 9.6% (50.0%) | 244.0% (NA) | 1.8% (0.0%) |

The Y chromosome is split into the Y pseudo-autosomal region (Y PAR; where heterozygous genotypes are allowed) and male-specific Y (MSY; where only hemizygous genotypes are allowed). The number of variants identified by the Ensembl Variant Effect Predictor as HIGH impact is given in parentheses.

**Fig. 2 | Structural variant calling in a long-read cohort. a** SV size distribution across the four SV classes (DEL deletion, INS insertion, INV inversion, DUP duplication) examined. Large spikes in insertions and deletions correspond to known transposable elements. **b** The majority of SVs which were only discovered through the long-read cohort compared to a pangenome panel have low minor allele frequency (AF), whereas those previously discovered are typically common SVs. Allele frequency was calculated from the cohort, and so the panel-only SVs have no assigned allele frequency. **c** Forced genotyping of SVs substantially reduces per-site missingness to similar levels identified in SNP variant calling. Markers represent the mean value and error bars indicate the 95% confidence intervals.

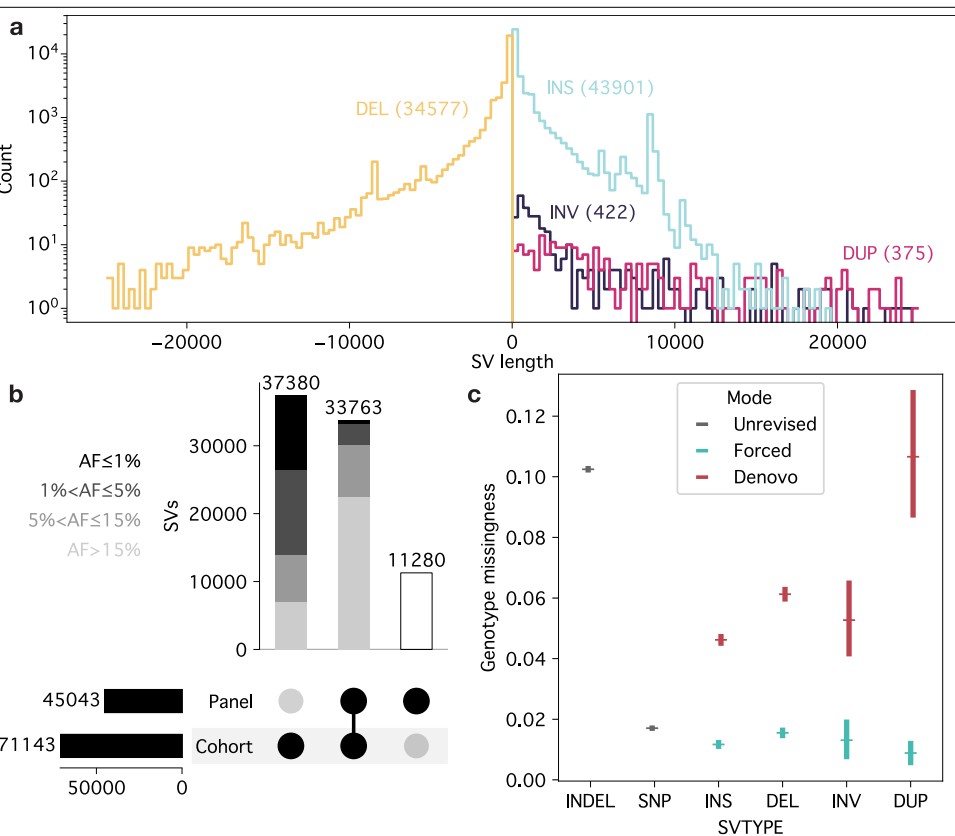

inspection of 15 HIGH-impact variants private to each group (Supplementary Table 1), we identified poor mapping of Illumina reads as the primary cause for the discrepancy between the variants called from short and long reads. All but one Illumina-only HIGH impact variant were likely genotyping artifacts as evidenced by implausible insert sizes and interchromosomal-mapping reads, with the sole exception of a singleton variant missed by the HiFi reads due to an uneven balance of alleles (7:1 read depth for the alleles). Conversely, all HiFi-only HIGH impact variants appeared correct, mostly missed by Illumina reads due to mapping quality of 0 or highly diverged regions potentially leading to local reference bias (Supplementary Fig. 4).

Previous evaluations suggest that, for small variants called from samples with moderate sequencing depth (8–15×), the misgenotyping rate may be as high as 2–5%[1,39,40]. For the present dataset, none of the 120 samples had microarray-called genotypes, thereby precluding a formal assessment of genotyping accuracy against a truth set. However, since previous research showed that DeepVariant calls cattle genotypes accurately from short reads[21], we considered the short-read variants as a truth and the HiFi variants as query. We found generally high genotyping concordance for SNPs on the autosomes (F1 = 0.93 ± 0.02), although there was a moderate drop (F1 = 0.86 ± 0.03) in genomic regions classified as tandem repeats (Fig. 1c). There was no such drop for regions containing transposable elements,

suggesting that there was sufficient variation within different transposable elements for short reads to correctly align, while they struggle in long tandem repeats with little motif variation. Variants genotyped on the X chromosome (F1 = 0.84 ± 0.04) and the PAR of the Y chromosome (F1 = 0.85 ± 0.02) were less concordant across all types of regions, while the limited number of short-read-based variant calls in the MSY region led to a substantial drop in F1 score (F1 = 0.27 ± 0.02). Indels behaved similarly, with an even more pronounced drop in accuracy around tandem repeats (Fig. 1d). Given the observed improvements in long-read alignments and more small variants called genome-wide, particularly in challenging regions, we only retained the HiFi-based small variants for downstream use.

We also called SVs from the HiFi read alignments with Sniffles2, finding 75,164 and 4111 SVs on the autosomes and sex chromosomes, respectively. We found approximately 23.9 ± 0.6 k SVs per genome, with only on the order of hundreds of additional SVs for each additional individual after 100 samples (Supplementary Fig. 5). This suggests that we likely captured the bulk of non-rare SVs present in the Braunvieh breed. SVs were overrepresented in regions annotated as tandem repeats, totalling just 3.2% of the genome, with 30.2% of SVs having at least half of their sequence contained within these regions. We observed a disproportionately large number of SVs of length ~300 bp, ~1.3 Kb, ~5.5 Kb, and ~8.4 Kb (Fig. 2a), corresponding to known transposable elements like Bov-A2 (SINE),

BTLTR1B (LTR), and BovB (LINE). Given the average read N50 of 16 Kb for this cohort, there is likely a small number of longer SVs that remains unidentified (Supplementary Fig. 6). We further found that 7944 SVs (10.02%) were present in only one sample (6988 singletons and 956 doubletons), ranging between 6 and 573 SVs unique to each sample. We found 37,380 additional autosomal SVs compared to a previous autosome-only, pangenome-derived SV panel constructed from one Piedmontese and 15 Braunvieh haplotypes not included in the HiFi-cohort analysed here, including 19,507 SVs with a minor allele frequency (MAF) between 1% and 15% (Fig. 2b), as well as 376 duplications and 422 inversions which cannot be easily genotyped through the previously applied $k$-mer approach[37]. The 11,280 SVs present only in the pangenome-panel are likely private to those animals or related to differences in assembly-versus-alignment-based SV calling.

During sniffles2 joint-calling, 'similar' SVs were merged into single alleles, collapsing multiallelic SVs. This overwhelmingly homogenised SVs that might only differ by small differences in length or position (Supplementary Fig. 7). Approximately 82% of the merged alleles had allele length standard deviations below 10, while only approximately 6% were above 100. However, SVs overlapping tandem repeats were particularly affected by merging alleles, either incorrectly treating different tandem repeat allele lengths as distinct SVs (rather than distinct alleles) or treating different tandem repeat allele lengths as a single consensus allele (Supplementary Fig. 8). Several large tandem repeats contained multiple SVs, primarily differing only by the start coordinate (corresponding to inserting/removing identical motifs at different locations within the tandemly duplicated sequence), which were not merged. On the other hand, smaller tandem repeats tended to be merged too aggressively, picking a consensus allele length which did not accurately reflect the copy-number diversity present in individual, pre-merged alleles.

Joint-calling SVs across the entire cohort resulted in a relatively high proportion of missing genotypes (mean of 5.3%) per-site, with duplications having the highest missingness (mean of 10.6%). Force-calling candidate SVs with sniffles2 from the original alignments substantially reduced the missingness to a mean of 1.3% (with the lowest mean missingness now observed in duplications at 0.9%), slightly lower than the missingness of joint-called SNPs (Fig. 2c). Genotypes that remained missing after force-calling were significantly associated with low alignment coverage ($p = 5.0 \times 10^{-23}$, Supplementary Fig. 9a) and, to a much lesser extent, mean read length ($p = 2.9 \times 10^{-6}$) after conducting a Type II ANOVA. Missingness was elevated at the start and end of chromosomes, roughly corresponding to centromeric and telomeric regions in the acrocentric cattle autosomes (Supplementary Fig. 9b), with other peaks corresponding to regions known to be highly polymorphic (e.g. BoLA on BTA23) or challenging for alignment (e.g. large segmental duplication on BTA10). Approximately 4% of the SV genotypes (380,911 genotypes across 54,561 unique SVs) changed during force-calling. The overwhelming majority of these changes (89.7%) were filled missing values (where there was potentially enough read support to force-genotype the SV despite not initially calling it), while 9.6% were changes between non-missing genotypes, and the remaining 0.7% changed from non-missing to missing genotypes (Supplementary Fig. 10a). A disproportionately high number of non-missing genotype changes were within centromeric-like regions (the first 200 Kb of chromosomes, corresponding to 0.2% of the genome), accounting for 11% of these unexpected changes. Other non-missing genotype changes occurred in regions where manual assignment of genotypes was not obvious from the alignments, with instances of force-calling either improving or worsening genotype accuracy (Supplementary Fig. 11). Force-called genotypes have been reported to be slightly less concordant than joint-called genotypes[41], although this negative consequence is very likely outweighed by the vast decrease in missingness.

Given the near-complete catalogue of SVs established for the cohort, we further examined the linkage disequilibrium between small variants and SVs. Even with the improved resolution of small variants within tandem repeats, we still found SVs overlapping tandem repeats as the most poorly tagged class of SVs. Over half of SVs (52%) containing tandem repeats were not in high linkage disequilibrium ($r^2 > 0.8$) with any small variant within a ±1 Mb window, compared to only 17% of SVs which did not contain tandem repeats or transposable elements.

The absence of a validated truth set of SV genotypes prevents a direct assessment of the SV genotyping accuracy achieved with Sniffles2. Nonetheless, the autosomal per-individual heterozygosity was significantly higher for small variant (23.99%) than SV genotypes called from the HiFi reads regardless if the de-novo (22.57%, Welch's $t$-test $p$-value = $3.45 \times 10^{-11}$) or force-called (21.96%, Welch's $t$-test $p$-value = $4.99 \times 10^{-19}$) SV genotypes were used (Supplementary Fig. 10b, c). The discrepancy between the per-individual small variant- and SV-based heterozygosity is less evident in six samples with more than 30-fold HiFi coverage (de-novo: 23.57% vs. 22.69%, Welch's $t$-test $p$-value = 0.07; force-called: 23.57% vs. 22.52%, Welch's $t$-test $p$-value = 0.04), suggesting that high sequencing depth is required to comprehensively call heterozygous genotypes with Sniffles2 and that heterozygous under-calling is more pronounced in force-called genotypes.

## molQTL mapping with HiFi data reveals the influence of structural variation on gene expression and splicing

The contribution of SVs detected from long-read sequencing to complex trait variability has not been investigated in large cohorts of cattle. To do so, we assessed impacts of small variants and SVs on molecular phenotypes through *cis*-molQTL mapping using deeply sequenced total RNA from testis tissue of the same bulls that was available from previous work[36]. After quality control on the RNA sequencing data, we considered 117 bulls for expression QTL (eQTL) and splicing QTL (sQTL) identification. Molecular phenotypes were estimated for 24,281 expressed genes and 16,975 spliced genes (corresponding to 50,853 intron clusters and 207,773 splice junctions), enabling association testing with 20,308,853 small variants (SNPs and small INDELs) and 61,861 SVs that were within 1 Mb of a molecular feature's start site and had a MAF of at least 1% and had non-missing genotypes in more than 90% of the samples (Table 2). The applied filtering substantially altered the characteristics of the variant set used for molQTL mapping, reducing the average per-site missingness to 1.21 ± 1.80 for small variants and 0.03 ± 1.07 for SVs. Among the variants included in the association analyses, 43.1% of small variants and 88.5% of SVs exhibited complete genotype data across all 117 samples.

Half of the expressed genes had at least one eQTL (12,584 genes; eGenes), corresponding to 16,644 independent-acting eQTL and 3,621,855 variants that passed the nominal significant threshold (eVariants; Table 2). This included 261 eQTL (218 eGenes) on the X chromosome and 15 eQTL (13 eGenes) on the Y chromosome. Approximately 40% of eGenes had at least one eVariant that was an SV. SVs were lead variants for 105 eQTL (hereafter referred to as SV eQTL; Table 2, (i.e. they either had the smallest p-value or the same *p*-value as the top small variant for an eGene. Three SVs were lead variants for multiple eGenes, including an 85,550 bp duplication on chromosome 8 (8:103486033–103571583) that was associated with *ORM1* and *COL27A1* expression. This SV was previously described as a likely causal variant for an increased expression of *ORM1* in liver tissue[26]. SV eQTL were depleted in intergenic regions and enriched in exons, CpG islands, promoters, and enhancers (Fig. 3a; Supplementary Table 2). We identified 23 SV eQTL that were not in strong LD ($r^2 \geq 0.8$) with a small variant, including a deletion that was associated with the expression of *ATXN7L3B* on chromosome 5. Notably, this 218 bp deletion was seven orders of magnitude more significant than the next eVariant (SV nominal p-value = $1.55 \times 10^{-17}$; Fig. 3b, c).

Nearly half of the alternatively spliced genes possessed sQTL (Table 2). We detected at least one sQTL for 10,388 intron clusters (23,292 splice junctions) within 7567 genes (sGenes), which totalled to 2,363,353 significant variants (sVariants) and 10,611 independent-acting sQTL. The X chromosome had 80 sQTL for 77 genes while the Y chromosome had 9 sQTL for 7 genes. Approximately 9% of the SVs considered for association testing were sVariants, totalling to 5535 SVs. SVs were lead variants (hereafter referred to as SVsQTL; Table 2) for 211 independent-acting sQTL

**Table 2 | Results for molecular QTL mapping with small variants and SVs**

| Variant type | Tested variants | eGenes | eQTL | eVariants(unique) | Top variants (unique) | sGenes | sQTL | sVariants(unique) | Top variants (unique) |
| --- | --- | --- | --- | --- | --- | --- | --- | --- | --- |
| Small variants | 20,308,853 | 12,584 | 16,664 | 5,866,009 (3,612,237) | 16,539 (16,334) | 7,567 | 10,611 | 6,363,647(2,357,818) | 10,400 (10,359) |
| SVs | 61,861 | | | 15,218 (9,618) | 105 (102) | | | 14,351 (5,535) | 211[a] (176) |

[a]136 SV sQTL contained an SV with the same p-value as the top small variant.
e/sGenes (gene that contains at least one eQTL or sQTL) and e/sQTL (independent-acting signals) include both small variants and SV. e/sVariants are defined as a variant that passes the gene's significance threshold, with the total number of unique variants listed in parentheses. Top variants are defined as the most significant variant for each independent-acting eQTL or sQTL, with the number of unique variants listed in parentheses.

for 173 genes and included 176 unique variants. One SV was associated with multiple intron clusters within the same gene, while 30 SVs were associated with multiple junctions from the same cluster. Only one SVs was associated with multiple sGenes; a 783 bp deletion on chromosome 25 (17,283,696 bp) was an SV sQTL for *VPS35L*, where the SV was located within an intron, and an sQTL for *KNOP1*, which was approximately 23 Kb upstream of the variant. SV sQTL were enriched in introns, exons, and splice sites, but depleted in intergenic regions (Supplementary Table 2). Approximately 16% of SV sQTL were not in strong LD with a small variant ($r^2 \geq 0.8$), though some of these SVs were much more significant than the next sVariant, such as a 514 bp deletion on chromosome 23 that was associated with alternative splicing of *ADGRF1* (Fig. 3d, e). One SV sQTL was not tagged ($r^2 < 0.2$) by any nearby small variants—a 120 bp insertion on chromosome 21 which was the only variant that passed the significance threshold for *LOC112443211*. The significant SV and gene both resided within a complex region near the beginning of the chromosome with over 150-fold coverage (at least 10 times higher than average) (Supplementary Fig. 12). The lack of LD with nearby SNPs is almost certainly driven by the SV and gene residing in a poorly resolved region of the reference genome, although the sQTL may still reflect a valid association.

We found that SV eQTL and SV sQTL exhibited similar characteristics. Specifically, low frequency variants had larger effect sizes (Fig. 3f, g), as did variants that were within the feature's boundary (gene for eQTL, intron cluster and gene for sQTL; Supplementary Fig. 13). Both eQTL and sQTL were underrepresented among low-frequency variants, reflecting the reduced statistical power to detect rare molecular QTLs (Fig. 3h). This underrepresentation was stronger for SVs than small variants. We observed no statistical difference in effect size between SVs and small variants for eQTL (Wilcoxon rank-sum test: $p = 0.23$; Fig. 3j), but insertions and deletions had slightly smaller effect sizes than small variants for sQTL (Wilcoxon rank-sum test: $p = 2.7 \times 10^{-8}$; Fig. 3j). Regardless, top variants for eQTL and sQTL were 2.1 (Fisher's exact test: $p = 9.29 \times 10^{-11}$) and 5.6-fold (Fisher's exact test: $p = 1.92 \times 10^{-70}$) enriched for SVs, respectively, highlighting the functional relevance of SVs. Manual inspection of molQTL regions and HiFi alignments confirmed that the SV top variants were genuine compelling causal candidates.

Approximately 55% of the SV molQTL contained the full or partial sequence of at least one transposable element (Fig. 3k). LINEs were the most common transposable element class among SV molQTL; however, they were underrepresented compared to their overall genome-wide abundance (Fig. 3l; Supplementary Table 3). LTRs were also depleted when considering their overall abundance, while DNA and RNA transposable elements were slightly enriched among SV molQTL, though not significantly (Supplementary Table 3). There was no statistical difference in effect size across transposable element classes for both SV molQTL; though, when considering classes with more than two observations, median effect size was slightly larger for RNA and LTR SV eQTL and LTR and SINE SV sQTL. Only 8% and 12% of SV eQTL and SV sQTL contained tandemly repeated motifs, which was lower than the overall proportion of SVs containing such repetitive sequence. However, we encountered cases where incorrectly merged multi-allelic tandem repeats were omitted as potential SV molQTL (Supplementary Fig. 14).

### Imperfect genotyping led to an underestimation of SV molQTL

Inspection of the molQTL summary statistics revealed many SVs that appeared as strong putative candidate causal variants but had slightly larger p-values than the top small variant. There were 167 eQTL and 168 sQTL where a small top variant was in perfect LD ($r^2 = 1.0$) with at least one SV that had a larger p-value (Fig. 4a, c). These differences in p-values across multiple pairs of variants in perfect LD were often driven by uneven distributions of missing genotypes. Because each variant within a pair of variants in LD exhibited its own pattern of missingness, the linear regression models were effectively fitted on different subsets of individuals. This slight mismatch in sample composition led to minor differences in estimated effect sizes and standard errors, and consequently in the resulting p-values.

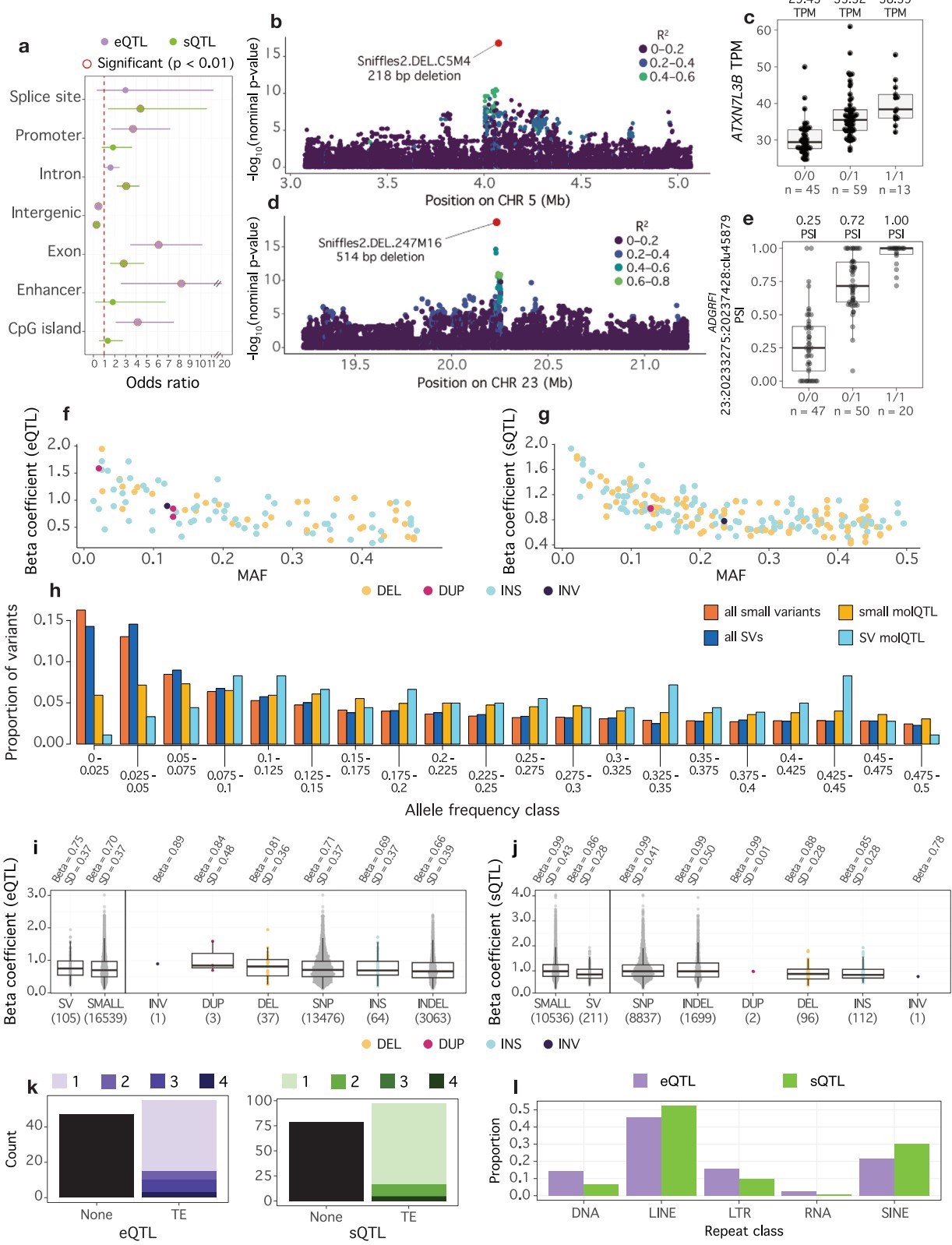

Nearly all (97%) of these 335 molQTL had at least one 'missing' call for the most significantly associated small variant. Only 17 of the 335 molQTL had a 'missing' call for the SV in LD.

The enrichment of SVs among molQTL was substantially stronger (6.1-fold enrichment for eQTL, Fisher's exact test: $p = 9.38 \times 10^{-130}$; 12.0-fold enrichment for sQTL, Fisher's exact test: $p = 8.04 \times 10^{-256}$) when

considering SVs in perfect LD with a top small variant. Furthermore, the proportion of SV eQTL and sQTL increased to 1.4% and 3.6%, respectively, when considering these molQTL. This included an *MAGI3* eQTL for which a SNP (3:29677355_A_G) was a lead variant. This SNP had a 'missing' genotype in one sample and was in perfect LD with a 6966 bp insertion that was within an intron of *MAGI3* but had non-missing genotypes for all

**Fig. 3 | SV molQTL characteristics. a** Enrichment of SV molQTL in functional elements. Odds ratios and *p*-values were inferred with a Fisher's test, with significant (*p* < 0.01) observations circled in red. **b** Manhattan plot for a poorly tagged SV eQTL that was associated with expression of ATXN7L3B. Variants are coloured by linkage ($r^2$) with the SV. **c** Boxplot of ATXN7L3 expression (TPM) across genotypes of the SV eQTL. Median TPM values for each genotype are reported above, and number of samples belonging to each genotype are reported below. **d** Manhattan plot for a poorly tagged SV sQTL that was associated with splicing of ADGRF1. **e** Boxplot of ADGRF1 intron usage (PSI) across genotypes of the SV sQTL. Median PSI values for each genotype are reported above, and number of samples belonging to each genotype is reported below. (f/g) Minor allele frequency (MAF) of each SV eQTL (**f**) and

sQTL (**g**), and effect size magnitude, coloured by variant type. **h** molQTL are depleted for rare variants, particularly for SVs. Colours correspond to variant type (SV or small variant) and whether the variant was a top variant. Effect size magnitude for eQTL (**i**) and sQTL (**j**) across variant types. Left panel shows all SV (≥50 bp) and small variant (<50 bp) molQTL, while the right panel is separated by specific variant types. Median effect size (Beta) and standard deviation (SD) is reported above, and number of variants within each category is below within parentheses. **k** Number of SV molQTL with transposable elements (TEs). Proportion is coloured by number of different TE classes for each SV eQTL or sQTL. **l** Proportion of TE SV molQTL across repeat classes.

**Fig. 4 | Misgenotyping of SVs impacts SV molQTL identification in 117 bulls.** Nominal *p*-value of top small variant and significant SVs for large effect eQTL (**a**) and sQTL (**c**). Colours correspond to whether the SV was in perfect linkage disequilibrium (LD) with the top small variant, had evidence of misgenotyping or if we did not identify an obvious error. Beta coefficient of the top small variant and significant SVs for large effect eQTL (**b**) and sQTL (**d**). Colours correspond to whether the SV was in perfect LD with the top small variant, had evidence of misgenotyping or if we did not identify an obvious error. **e** Comparing sample-wise average coverage across chromosome 2 and coverage of the 681,722 bp duplication clearly separates heterozygous carriers of the duplication (black symbols) from non-carriers (grey symbols). The red dotted line is an identity line. As expected, heterozygotes demonstrated on average a 1.51-fold (between 1.43 and 1.62-fold) coverage increase across the duplication relative to chromosome 2 background coverage. Four heterozygous samples (yellow stars) were misgenotyped as homozygous reference by Sniffles2. **f** Boxplots representing the expression of STK39 and CERS6 in carriers (0/1) and non-carries (0/0) of the 681,722 bp duplication the was misgenotyped for 4 samples (red symbols). Median TPM values for each group are listed above.

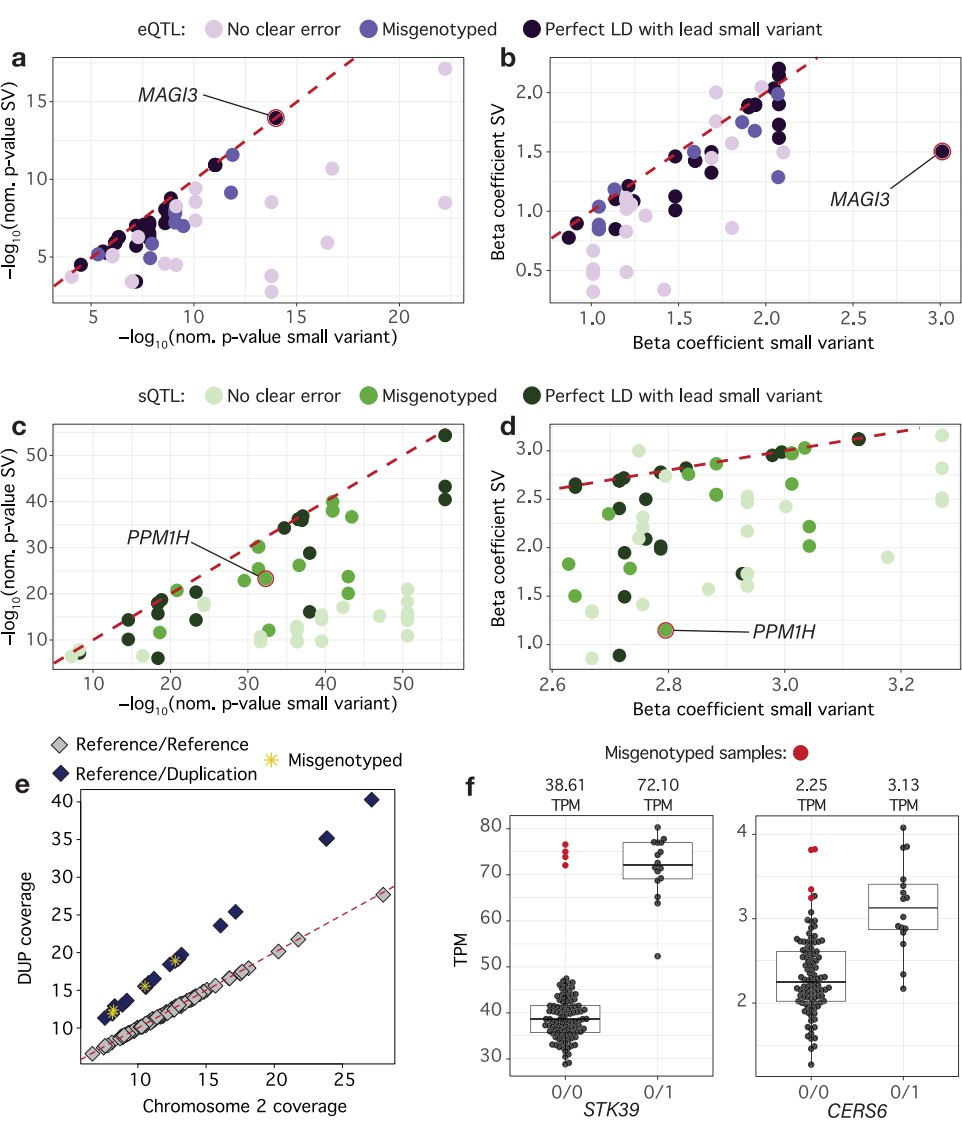

samples. The nominal *p*-value of the SNP (*p*-value = $1.03 \times 10^{-14}$) was slightly lower than that of the SV (nominal *p*-value = $1.14 \times 10^{-14}$; Supplementary Fig. 15), demonstrating that an unequal number of samples with missing genotypes across variants introduces bias in association tests. We also identified instances where 'missing' genotypes for SVs resulted in missed SV molQTL. For instance, we observed an 8790 bp insertion that had 'missing' genotypes in four samples, which we identified as homozygous for the reference allele upon manual inspection of the overlapping read alignments (Supplementary Fig. 16). This insertion was in complete LD with a small variant sQTL for *LOC768028*. Exclusion of the 'missed' genotype samples for this SV during association testing had a substantial impact on

the estimated p-value (nominal *p*-value = $4.97 \times 10^{-12}$), which was consequently fifteen orders of magnitude larger than the *p*-value of the top SNP (nominal *p*-value = $3.60 \times 10^{-27}$).

The effect size distributions for both sQTL and eQTL revealed that the most extreme molecular QTL effects were primarily driven by small variants rather than SVs (Fig. 3h, i). We manually examined the top 1% of large effect molQTL with at least one SV passing the gene's significance threshold to determine whether erroneously genotyped SVs prevented their identification as molQTL. Among the 33 eQTL and 31 sQTL that passed these criteria, we identified genotyping errors for SVs that were strong causal candidates for 9 eQTL and 10 sQTL (27.3% and 32.3% of the manually

examined large effect eQTL and sQTL, respectively; Fig. 4a–d). Thus, while nearly a third of these variants were affected by genotyping errors, the extent of misgenotyping per variant was low; for most of the manually examined SVs, only one or two samples showed genotyping errors, corresponding to a per-variant error rate of approximately 2%. Most genotyping errors stemmed from low overall coverage, reads failing to span the length of the variant, and insufficient coverage of one haplotype (often leading to heterozygous genotypes being miscalled as homozygous; Supplementary Fig. 17). Large insertions were particularly susceptible to misgenotyping, and we observed multiple instances of insertions exceeding 8 Kb being missed as SV molQTL. Notably, the set of eVariants and sVariants was depleted for insertions between 8 and 9 Kb, highlighting the broader effects of misgenotyping variants of this size (Fisher's exact test: eVariants: OR = 0.41, $p = 5.82 \times 10^{-18}$; sVariants: OR = 0.27, $p = 1.08 \times 10^{-18}$). For example, we identified an 8573 bp insertion that contained an L1 LINE element and was associated with an alternative splicing event of *PPM1H*. This variant was nearly ten orders of magnitude less significant than the top SNP (SV nominal $p$-value = $5.20 \times 10^{-24}$; SNP nominal $p$-value = $5.13 \times 10^{-33}$; Fig. 4c, d), despite being located within the intron that was affected by differential splicing. Inspection of the HiFi alignments revealed that two samples were misgenotyped as homozygous insertion carriers due to low coverage of the reference haplotype (Supplementary Fig. 18). Correcting the two misgenotyped samples resulted in a stronger association between the splicing event and the SV (corrected SV nominal $p$-value = $1.31 \times 10^{-26}$).

Our SV catalogue contained 43 duplications spanning from 3 to 956 Kb that completely overlapped 135 testis-expressed genes, yet only four were identified as molQTL. This low proportion is unexpected, considering that previous studies have highlighted the substantial impact of such duplications on gene expression[26]. Nineteen of these 43 duplications could be confidently verified though coverage-based genotyping, corroborating that sniffles can identify duplications that are much larger than the HiFi reads. Taking the coverage-based genotypes as truth and the sniffles-called genotypes as query, we found that at least one sample was misgenotyped for all nineteen variants examined. Overall, we found that a substantial fraction of the sniffles-called genotypes for these long duplications were erroneous in both lowly and highly covered samples (Supplementary Fig. 19). We identified a relatively common 682 Kb duplication on chromosome 2 (2:27313972–27995694), which overlaps the entire coding sequences of *CERS6* and *STK39*. This duplication was called as heterozygous by Sniffles in 16 samples; however, the coverage-based genotyping indicated that 4 heterozygous samples were incorrectly genotyped as homozygous reference (i.e. did not have the duplication; Fig. 4e; Supplementary Fig. 20), despite having moderate coverage. These genotyping errors resulted in this duplication being missed as an SV eQTL for both overlapped genes. We observed that the top small variants for these genes were one to three orders of magnitude more significantly associated than the erroneously genotyped duplication (top small variant $p$-value *CERS6* = $3.31 \times 10^{-15}$, *STK39* = $1.90 \times 10^{-14}$; duplication $p$-value *CERS6* = $8.34 \times 10^{-12}$, *STK39* = $7.17 \times 10^{-13}$; Fig. 4f). However, if genotyped correctly, the duplication is the top variant for both genes (corrected duplication $p$-value *CERS6* = $1.33 \times 10^{-15}$, *STK39* = $3.96 \times 10^{-15}$).

## Discussion

We have generated and analysed a cohort of Braunvieh cattle that exceeded the effective population size with long and accurate HiFi reads, representing one of the largest long-read datasets available for a non-model organism. Notably, this bovine long-read DNA cohort also has paired functional data, providing the opportunity to investigate the consequences of small and structural variation on molecular phenotypes. This cohort enabled us to detect and characterise 23,789,890 small variants and 79,275 SVs, including tens of thousands of previously unidentified SVs. Our collection of SVs is almost twice as large as that identified in a previously built 16-sample cattle pangenome[37] and substantially larger than previously reported in similar-sized, or larger, short-read sequenced cattle cohorts[25–27]. While short-read based SV cohorts are either depleted for insertions or lack them

completely[9,25,42], we identified—as expected[23,43]—more insertions than deletions from the long-read alignments, thereby providing comprehensive access to a vastly understudied variant type. A saturation analysis corroborated that our SV set is a near complete catalogue of common SVs segregating within the Braunvieh population. However, we suspect that longer insertions and duplications (>15,000 bp) were missed for samples with shorter average read lengths. Long-read sequence data also improved alignments to the X and Y chromosomes—which are often neglected in GWAS for several reasons, including poor genotyping resulting from alignment errors due to their repetitiveness—and thereby facilitated the identification of over one hundred thousand small variants that were previously inaccessible with short-read sequencing, including 7 SNPs predicted as HIGH impact. However, variants along the MSY remained poorly characterised due to uneven coverage and copy-number complexity, limiting the ability to identify HIGH impact variants and molQTL for genes within this region. We also demonstrated improved small variant calling on the autosomes, particularly in tandem repeat regions, when using long reads over short reads, thereby creating a comprehensive genetic resource for genome-wide association testing.

Our expression and splicing QTL analyses revealed that molQTL are enriched for SVs. Despite being derived from a specific phenotype (testis RNA sequencing data), our findings support the broader importance of SVs in complex trait variation, as previously reported in cattle and other species[15,44–46]. We observed that 0.6% of eQTL and 1.9% of sQTL had an SV as the top variant. These proportions of SV molQTL were between 2.1-times and 14-times greater than those reported in short-read-based studies in bulls[25,37]. More than half of the SV molQTL were due to insertions that were largely inaccessible with short reads. Notably, the number of the SV molQTL nearly doubled when accounting for SVs in complete LD with a lead small variant, increasing to 1.4% of eQTL and 3.6% of sQTL. These estimates are more comparable to short-read- and long-read-based SV molQTL studies that were conducted in humans with higher coverage sequence data and larger sample sizes[46,47]. Nonetheless, confirming a causal role for these SVs requires further functional experiments. Contrary to other SV molQTL studies, we did not observe a statistical difference in effect size magnitude between small variants and SVs[46]. However, we identified multiple large-effect molQTL where small variants were prioritised as candidate causal variants despite the presence of compelling candidate causal SVs. We suspect that several large allelic substitution effects were erroneously attributed to the small variants due to imperfect SV and small variant genotyping, which introduces some bias in this assessment.

The forced genotyping approach substantially reduced the proportion of missing genotypes for all types of SVs. Yet, many SVs were missed as molQTL due to erroneous SV genotypes and an unequal distribution of missing genotypes between SVs and small variants. Errors in SV genotyping can have a substantial impact on QTL detection; for example, Chiang et al.[46] reported that a 5% increase in error rate reduced SV eQTL detection by 19.6%. Almost one-third of the manually examined molQTL and all manually examined large duplications were affected by imperfectly genotyped small variants and SVs. The per-variant error rate among the manually examined molQTL was relatively low. However, because our sample size was small, even a few misgenotyped samples could strongly influence molQTL detection. We also observed instances, e.g. for large duplications, where much higher per-variant error rates obscured SV molQTL detection. Using the manually revised genotypes, we identified compelling candidate causal SVs in many of the examined molQTL where the summary statistics from the association tests initially prioritised small variants. We were able to attribute some of the SV genotyping errors to insufficient coverage and the undercalling of heterozygotes. This suggests that an average HiFi coverage higher than the 13.5-fold obtained for our cohort will produce more accurate genotypes for both small variants and SVs, which will benefit association testing. Such findings also indicate that SVs are even stronger enriched among molQTLs than reflected by the results from the statistical association testing. Imputation can enhance statistical power and mitigate some of the missing genotype biases in

association testing that we observed[48,49]. However, the accuracy of imputation can be much lower for SVs than small variants, which negatively impacts association testing and causal variant identification[50]. Since it remains unclear whether the benefits outweigh the drawbacks, we did not attempt to impute missing genotypes, neither for SVs nor for small variants. Generating high-coverage long-read data at the sample sizes required for detecting QTL is becoming increasingly feasible[17,51], which will reduce the missingness, thereby improving the statistical power of association studies utilising long-reads for variant discovery and genotyping. In the meantime, moderate-coverage long-read sequencing data, when collected from cohorts that adequately reflect the haplotype diversity of the target population as demonstrated here, will enable the development of population-scale analytical methods such as SV genotype imputation[52].

Genotyping errors within complex regions, such as those containing multiallelic SVs or tandem repeats, and an inappropriate merging of SVs within these regions likely caused the observed underrepresentation of SV molQTL annotated as tandem repeats. In addition, the lack of reads spanning large variants resulted in erroneously genotyped large SVs, including some molecular phenotype-associated variants, which remained undetected by our association analyses. This contributed to the lack of association of gene-containing duplications and likely contributed to the observed underrepresentation of LINE and LTR transposable element molQTL, which have previously been associated with large impacts on gene expression[53,54]. Further research is warranted to develop refined genotyping and association testing methods to fully capitalise on long-read sequencing data and explore the association between functionally relevant SVs and complex traits[55].

## Methods

### Ethics statement
The *Bos taurus taurus* tissues used in this study were previously collected from a commercial abattoir in Zürich, Switzerland after regular slaughter. The slaughter procedures at the abattoir were in line with government regulations. Bulls were slaughtered using a two-step process consisting of captive-bolt stunning followed by severing the major blood vessels in the neck. Following slaughter, the scrotum was removed and transported to the laboratory for further preparation. None of the authors were involved in the decision to slaughter the bulls. No ethics approval was required for this study.

### Sample extraction and DNA sequencing
Reproductive tissues (including testis and the caput of the epididymis) from 120 mature bulls were sampled by Mapel et al.[36] from slaughtered bulls within 40 min to 270 min after slaughter. Tissue samples were flash frozen in liquid nitrogen and stored at −80 °C. Sampling was conducted between 2019 and 2021 and HiFi sequencing was performed from December 2022 to May 2023.

We extracted HMW DNA from flash-frozen testis ($N = 105$) or epididymis ($N = 15$) tissue samples with the Monarch HMW DNA Extraction Kit for Tissue (New England Biolabs). We followed the manufacturer's recommended protocol for both tissue types. DNA was shipped on dry ice to PacBio (Rolling Stock Yard, London) for fragment length analysis, library preparation, and sequencing.

### DNA read alignment
We aligned the HiFi reads with minimap2 (v2.24)[56] to ARS-UCD2.0 with the X chromosomal PAR hard masked (X:133300518-139009144), using the 'map-hifi' preset. Alignments were coordinate-sorted with SAMtools (v1.19.2)[57]. The previously collected short reads were aligned with bwa-mem2 (v2.2.1)[58,59] to the same reference, before collating by name, marking duplicates, and coordinate sorting with SAMtools.

We assessed alignment coverage using SAMtools bedcov and BEDtools (v2.30)[60] coverage, using 100 Kb windows generated by BEDtools makewindows. Regions with a minimum primary read depth of 2 and MAPQ of 5 were considered suitable for variant calling.

### Small variant calling
We called small variants with DeepVariant (v1.6.0)[61], using the 'PACBIO' model for the HiFi reads and 'WGS' for the short Illumina paired-end reads, specifying—haploid_contigs 'X,Y' and the PAR bed file, producing a gvcf for each sample. Samples were merged into population vcfs using GLnexus (v1.4.1)[61] with the DeepVariant config and revise_genotypes=False. Deep-Variant variants larger than 45 bp were removed due to avoid unintentional overlapping with the dedicated SV calling.

We classified regions as tandem repeats using the pbsv (v2.9.0) utility findTandemRepeats (https://github.com/PacificBiosciences/pbsv/blob/master/annotations/findTandemRepeats) or as transposable elements using the RepeatMasker (v.4.1.5) (https://www.repeatmasker.org/) utility RM2Bed.py on the UCSC browser repeat file (https://hgdownload.soe.ucsc.edu/hubs/GCF/002/263/795/GCF_002263795.3/GCF_002263795.3.repeatMasker.out.gz) and keeping all SINE/LINE/LTR elements. Regions overlapping in tandem repeats and transposable elements were prioritised into the tandem repeat bed. All remaining regions were assigned as "normal" using BEDtools complement.

We assessed variant calling and genotyping accuracy per sample with hap.py (v0.3.15), removing all reference calls, left-shifting indels, and stratifying with the genomic regions specified above.

### Structural variant calling
We called SVs with sniffles2 (v2.2)[62], providing the tandem repeat locations from above. We merged per-sample snf files across the cohort with Sniffles2, removing any breakend SVs or SVs larger than 1 Mb, before force-genotyping each sample again with the merged set of SV candidates. We finally merged force-called VCFs with BCFtools merge and removed SVs with more than 10% missingness.

We assessed the repetitive element and tandem repeat content of the SV sequence (the reference for deletions and alternative for insertions) using Repeatmasker and TRF (v4.09.1)[63], respectively.

### Variant analyses
We used VEP (release 113)[64] to assess functional impacts and consequences of small variants using the RefSeq (release 106) cattle annotation. We used plink (v1.90b6.26)[65] to calculate the LD between SVs and small variants, using a 1 Mb window and a minimum $r^2$-threshold of 0.2 and 0.8 to assess weak and strong tagging, respectively. We tested for mutually present SVs between our cohort and the SV pangenome panel from Leonard et al.[37] using Jasmine (v1.1.5)[66], with the flags "--pre_normalize max_dist_linear=0.5 max_dist=250" to allow slight mismatches in SV position or length between the two callsets. We used BEDtools intersect to examine the overlap between SVs and the previously described tandem repeat file.

### Coverage analysis
We used mosdepth (version 0.2.2)[67] to calculate the average coverage in 500 bp windows for the HiFi alignments. Heterozygous and homozygous genotypes were assigned to the sniffles2-called duplications when the average coverage over the duplicated sequence was at least 1.3-fold and 1.8-fold higher, respectively, than the average coverage outside the duplication on the chromosome containing the duplication.

### Molecular phenotype preparation
Total RNA from the HiFi-sequenced individuals was sequenced previously and made publicly available in Mapel et al.[36] (Supplementary Data 1). We filtered RNA reads with fastp (v0.23.1)[68] to remove adaptor sequences, poly-A-tails, poly-G-tails, and low-quality bases.

We used Kallisto (v0.50.0)[69] and the RefSeq annotation for cattle to quantify transcript expression (TPM) and counts, which were aggregated to gene-level with tximport (v1.34.0)[70]. We considered genes with TPM ≥ 0.1 and at least 6 supporting reads in ≥10% of samples for eQTL mapping. Expression values were inverse normal transformed and quantile normalised.

**Article**

To identify and quantify splicing events, we first aligned cleaned reads to the ARS-UCD2.0 reference and RefSeq annotation with STAR (version 2.7.11a)[71] and included WASP filtering[72] to account for allelic bias, using heterozygous sites called from the HiFi reads. We extracted exon-exon junctions with Regtools (v0.5.2)[73], then used LeafCutter (v0.2.9)[74] to cluster introns, calculate intron excision ratios, perform filtering, and normalise splicing phenotypes for sQTL mapping.

## molQTL mapping

Cis-molQTL mapping was conducted with QTLtools (v1.3.1)[75]. We included variants that were within a 1 Mb window of the feature's start site (gene for eQTL and intron cluster for sQTL), and had MAF ≥ 1%, and missingness <10%. We selected covariates for eQTL and sQTL testing with PCAForQTL[76] which implemented an elbow test to estimate the number of principal components (PCs) from the expression or splicing matrix to include as hidden confounders (hereafter, 'RNA PCs'). We used the filter-KnownCovariates function ($R^2$ thresholds = 0.5) to remove known covariates that were captured by the RNA PCs. The covariates considered for eQTL mapping included PCs 2–10 of the genome relationship matrix (constructed from 381,111 unlinked variants with plink (v2.00a3.6LM)[65] and --indep-pairwise 1000 5 0.2, RIN, age, and RNA PCs 1-12. The sQTL covariates included PCs 2–10 of the genome relationship matrix, RIN, age, and RNA PCs 1–11. We conducted 1000 permutations with --permute function in QTLtools to infer beta-corrected p-values, then used the qtltools_runFDR_cis.R script to apply a 5% false discovery rate (FDR) and estimate significance thresholds for each gene. sQTL testing includes the --grp-best flag to account for multiple intron clusters within a gene. We performed a conditional analysis using the --permute and --mapping flags of QTLtools[75] to identify independent-acting signals for each gene or intron cluster and the corresponding most significant variant.

We defined "SV molQTL" as a QTL for which either an SV had the smallest p-value, or an SV had a p-value that was identical with a lead small variant. Enrichment was inferred with a Fisher's exact test. We used BED-tools to identify SVs that overlapped functional elements. Promoters and enhancer annotations for cattle testis tissue were obtained from Salavati et al.[77], while CpG islands were obtained from the UCSC genome browser[78]. Abbreviations and full descriptions of all discussed genes are listed in Supplementary Table 4.

## Statistics and reproducibility

Bulls of Braunvieh ancestry were randomly sampled at a slaughterhouse without prior knowledge on their exact age and origin. Sample size was primarily defined based on available resources. RNA and DNA extraction and sequencing was conducted in batches, and all data were analysed together upon completion of data acquisition. Results are expressed as mean ± standard deviation (SD) where appropriate. Covariates inferred from the RNA and DNA sequencing data and from the sample metadata were used to account for possible confounding factors in the molQTL mapping experiments. A false-discovery rate of 5% was applied to beta-corrected p-values that were inferred from permutation testing to identify molQTL.

## Reporting summary

Further information on research design is available in the Nature Portfolio Reporting Summary linked to this article.

## Data availability

DNA and RNA sequencing data of the analysed cohort are available in the ENA database at the study accessions PRJEB42335 (Long-read sequencing data from cattle for the purpose of de-novo genome assembly), PRJEB28191 (Short read sequencing of cattle) and PRJEB46995 (Testis transcriptome of mature bulls). Accession identifiers for all samples are available as Supplementary Data 1. Gene expression and splicing matrices, a VCF file of genome-wide small and structural variant genotypes used for e/sQTL mapping, a cross-table to link genotype and transcriptome data as well as

results from e/sQTL mapping have been archived at zenodo (https://zenodo.org/records/15431126)[79]. The source data behind the graphs in the paper can be found in Supplementary Data 2 and at Zenodo (https://zenodo.org/records/15431126)[79].

## Code availability

Computational workflows are available through https://github.com/AnimalGenomicsETH/HiFi_cohort and are archived at zenodo (https://zenodo.org/records/18172395)[80].

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

## Acknowledgements

This study was supported by an ETH Research Grant and a grant from the Swiss National Science Foundation (SNSF, grant-ID 204654). The funding bodies were neither involved in the design of the study and collection, analysis, and interpretation of data nor in writing the manuscript. We thank Eirini Lampraki from Pacific Biosciences for DNA fragment analysis and sequencing. We thank Audald Lloret-Villas and Qiongyu He for valuable discussions.

## Author contributions

X.M.M. sampled tissue and purified HMW DNA, aligned RNA reads against the reference, developed and applied workflows to quantify gene expression and splicing variation, conducted molecular QTL mapping, interpreted results, and drafted the manuscript; A.S.L. aligned DNA reads against the reference, called variants from short- and long-read alignments, interpreted results, and drafted the manuscript; H.P. conceived the study, contributed to the analysis of expression and splicing QTL, interpreted results, and contributed to the writing of the manuscript. All authors approved the final version of the manuscript.

## Competing interests

The authors declare no competing interests.
