## [Transparent Peer Review file · Communications Biology]

Molecular QTL are enriched for structural variants in a cattle long read cohort

Corresponding Author: Professor Hubert Pausch

Version 0:

Reviewer comments:

Reviewer #1

(Remarks to the Author)

The manuscript by Mapel et al. describes an analysis, in cattle, that investigates the impact of structural variations on molecular phenotypes using a QTL approach. The work involved the production of long reads HiFi sequences for 120 bulls and the detection of small variants as well as structural variants from the alignment of those reads against the ARS-UCD2.0 genome assembly. Both small variants and structural variants (SV) are used to detect quantitative trait linkage with gene expression and gene splicing, an approach called molQTL. The main results of the paper include comparative analysis of short (Illumina) and long (HiFi) reads technology for small variant discovery, the added value of force-calling approach for genotyping structural variants in comparison to the genotypes provided by the Sniffles software, and, what is probably the main result of the paper, enrichment in structural variants of the top eQTL and sQTL variants. The analyses presented were conducted using appropriate methods, and the results shed light on the contribution of structural variation to molecular phenotypic variability in cattle. However, some aspects of the analyses performed and the way the results are presented raise several concerns.

1. The result section starts with a long description (almost 3 pages including figure 1) of a comparative analysis of short versus long read sequences for the purpose of detecting small variants. This question has been thoroughly addressed, using the same technologies (Illumina and HiFi) in the paper by Kosugi and Terao (ref 9 in the manuscript). The authors should explain the added value of their analysis and refer to the paper of Kosugi and Terao to explain the added value.
2. The scatter plot (Figure 1a) describing the relationship between short and long reads sequencing depth does not make sense in our point of view, coverage depth is a choice made when generating the data. The information provided by the plot is that some samples have an HiFi coverage significantly higher than the Illumina coverage. Difference in coverage between samples, and its impact on variant detection, is however not addressed in the analysis.
3. By addressing specifically “high impact variant” as predicted by Ensembl Variant Effect Predictor (EVEP), the authors imply that these variants may have specific characteristics that influence (short versus long read) variant detection. If that is indeed the case, the authors should explain why.
4. The authors mention an “observed improvements in small variant genotyping from long reads” that, to the best of our understanding, is not quantified nor really supported by the presented analysis (last sentence before addressing the detection of SV in the result section). An increased number of variants is observed in chromosome X, and 15 high impact variants not shared by both technologies seem to be associated with poor mapping of illumina reads but this does not imply an observed improvement in small variant genotyping on the large set of detected variants (~20 millions).
5. The second analysis presented in the results section deals with structural variant detection. Here, in contrast, no comparative analysis with SV detection using short reads is presented. We understand that this is not the focus of this work, but highlighting the added value of long reads for SV detection seems important in a study emphasizing the added value of considering SV detected by long reads to molQTL detection. The authors have in hand a unique dataset of 120 bulls sequenced with both short Illumina reads and long HiFi reads. Including a comparison of the variant catalogs from the two technologies on a dataset made on a large number of individuals would certainly greatly enhance the paper. The authors have however already addressed this question in reference 19. A correspondence between the two studies could be made by the authors in order to enable readers to fully benefit from the results presented in both studies. A better balance between

the detection of small variants and the detection of structural variants would be desirable.

6. The authors implicitly suggest a relationship between the effective population size of the Braunvieh breed (70) and the fact that from the 100th sample onwards, approximately 100 new SVs are observed per additional sample (in Supplementary Figure 5 they even specify that this occurs starting from the 88th sample onwards). For a reader not familiar with population genetics, the connection between the two numbers is difficult to understand (70 and 100 or 70 and 88). If the relationship is simple it should be explained or a reference could be given. If the relationship is not straightforward it is probably better to omit the precise value of the effective population size to avoid disrupting the reading.

7. It is mentioned that 37,380 novel autosomal SVs were found compared to a previous pangenome-derived panel (constructed from 15 (?) Braunvieh haplotypes). Reference 19 should probably be mentioned here (in ref 19, a pangenome based on 16 assemblies is mentioned) The relationship between the two populations (ref 19 and this work) is not explained. We believe that some variants found in the pangenome-based approach were missed by the present one. This could be simply mentioned.

8. The authors underline the benefits of force-calling to obtain reliable genotypes. It should probably more clearly stated in the text that force-calling is a feature offered by Sniffles.

9. The fact that "Force-calling SVs with sniffles2 overwhelmingly filled in sporadically missing genotypes" is, to the best of our understanding, not a surprise since the corresponding SVs were not found in the associated samples and could therefore not be genotyped (there is no corresponding line in the vcf file of those samples). If this is the case, this should be explained in the text by the authors.

10. The precise meaning of unique variants, top variants and associated unique variants is not explained (Table 2). This should be stated explicitly to prevent the reader from having to infer it or search the literature.

11. molQTL detection with small variants and SV in Braunvieh were already addressed in a previous study by the authors (ref 19). A correspondence between the two studies should be made by the authors in order to enable readers to fully benefit from the results presented in both studies.

12. What do the authors mean by "feature's significance threshold" ? (penultimate paragraph of the results section).

13. In Figure 4e, the black diamonds seem to exhibit a linear relationship which is not discussed in the text.

14. The impact of genotyping - both structural variants (SVs) and small variants - on the ability to detect molQTLs is a very interesting point, and directing readers to relevant literature on this topic would be highly valuable.

15. In the discussion, the authors indicate that they suspect long SVs (>15,000 bp) are missed, given the HiFi read lengths. The authors could mention that this applies to insertions (and duplications) and not to all SV (large deletions are detected as shown in Figure 2).

16. Is the X chromosome really neglected in GWAS studies due to its repetitiveness?

17. The ENA ERP126174 project (last updated in 2021) contains 24 runs (some of which are Illumina reads) and is already mentioned in ref19 and does not contain the Hi-C reads analyzed in the article. The HiFi reads from the 120 bulls should be made publicly available.

The work presented here constitutes a contribution to the understanding of the impact of structural variation on gene expression and splicing. We believe that the paper would greatly benefit from a rewrite of the Results section, placing less emphasis on raw numbers (such as the number of SVs, percentages, and p-values) and instead focusing more on the questions addressed and how the data help to answer them.

Reviewer #2

(Remarks to the Author)

This manuscript by Mapel et al. presents a comprehensive analysis of structural variants (SVs) and their impact on molecular phenotypes in cattle using long-read HiFi sequencing. The authors sequenced 120 *Bos taurus taurus* bulls of primarily Braunvieh ancestry using PacBio HiFi technology, generating 4.86 terabases of data with an average read N50 of 16.3 kb at 13.5-fold coverage. This represents one of the largest long-read datasets available for a non-model organism and addresses a critical gap in cattle genomics where previous SV studies have relied primarily on short-read sequencing, pangenome analyses, or QTL fine-mapping approaches.

The study's strength lies in its integrative approach, combining comprehensive variant discovery with functional characterization through molecular QTL (molQTL) mapping using deeply sequenced total RNA from testis tissue of the same individuals. This design enables direct assessment of SV impacts on gene expression and splicing variation, moving beyond mere variant cataloging to functional interpretation.

Overall Assessment

The manuscript is very well-written and logically organized, with clear figures and comprehensive supplementary materials. The methods section provides sufficient detail for reproduction, and the results are presented systematically. However, I have several major points for the author team to address/clarify before accepting the MS.

Major Comments

- Force-calling substantially improved missing data rates, but you report that 4% of genotypes changed during this process. How did you validate that these changes represent improvements rather than the introduction of new errors?
- The reported 27–32% misgenotyping rate is derived from the top 1% largest-effect molQTL, I have a feeling this is only valid as a “worst-case” indicator in the most error-prone, high-effect loci, but it likely overestimates the genome-wide rate.
- Could the authors comment on their approach to assign heterozygous and homozygous genotypes for duplications? Fixed fold-change cutoffs might have their own caveats. In addition, could the author provide their thoughts on duplications and how well we can use them to predict impact?
- Is it possible re-estimate SV vs SNP effect sizes using a high-confidence SV subset (e.g., non-TR loci, size < 10 kb, strong read support and breakpoint consistency). This will help determine whether the observed effect-size equivalence is driven by genotyping error.
- Based on your experience with genotyping errors and coverage limitations, what specific recommendations would you make for future long-read SV studies in livestock?
- Given the focus on male fertility traits, what is the practical significance of your findings for cattle breeding? Can you quantify the potential economic impact of incorporating SV information into genomic selection programs?
- Figure quality: Some supplementary figures have poor legends and inconsistent formatting (detailed in minor points below).

The absence of line numbers makes it very difficult to specify where corrections are needed. Below are some minor points.

Minor Points

- Consider creating an abbreviation table for all gene names (e.g., STK39, MAGI3, CERS6).
- Scan the manuscript again for legend abbreviations/annotations, several times I can find figure has abbreviations but there was no annotation in the legends (e.g., VNTR, MSY).
- “We identified 23 SV eQTL that were not in strong LD ($r^2 \geq 0.8$)” – For some reason I found several r^2 superscripts are hyperlinked to citations.
- Typo: “ployp-G-tails” → “poly-G-tails”
- Please maintain consistent mathematical formatting:
“We considered genes with $TPM \geq 0.1$ and at least 6 supporting reads in $\geq 10\%$ of samples for eQTL mapping.”
- Keep consistent with “long-read” (with or without hyphen).
- Some references are inconsistently formatted:
Nat Commun vs. Nature Communications
- Abstract: “Sequencing mapping cohorts” → “Sequencing cohorts”
- Add a short cautionary note regarding interpretability of MSY molQTL given uneven coverage and copy-number complexity.
- Introduction: “orders of magnitudes longer” → “orders of magnitude longer”
- Results: “on the order of 100s” → “on the order of hundreds”
- Supplementary Figure 1: Reports “Pearson’s $r^2=0.65$, $p=5.9e-37$ ” while main paper states “Pearson’s $r: -0.80$, $p=5.87 \times 10^{-37}$ ” – which is correct?

Reviewer #3

(Remarks to the Author)

Mapel et al. present a comprehensive study leveraging HiFi long-read sequencing in 120 Braunvieh bulls to generate an exhaustive catalog of both small variants (23.8 M) and structural variants (SVs; 79.3 k) in cattle testes. Combining these genomic data with total RNA-seq from testis tissue of 117 samples, the authors performed cis-molecular QTL (molQTL) mapping for expression (eQTL) and splicing (sQTL). The present study offered a huge amount of data and candidate loci contributed to complex traits. However, there are still some questions required to be addressed:

All molQTL analyses are derived exclusively from testis RNA. Regulatory landscapes differ markedly across tissues (e.g. liver, muscle, mammary gland), so conclusions about SV impacts on “complex traits” may not extrapolate to traits governed by other tissues. Additional RNA-seq from diverse tissues would strengthen claims of SV enrichment genome-wide.

SVs prioritized as putative causal variants (e.g. ATXN7L3B deletion, ADGRF1 splice-junction deletion) are identified solely through statistical association and manual inspection. Functional validation (e.g. reporter assays, CRISPR perturbation) is needed to confirm their regulatory effects. Without such validation, mechanistic claims remain speculative. If this is impossible at this stage, maybe discuss it in the discussion part.

What are small variants in the abstract, SNP? InDels or other else?

What does the author mean about “the top variants” in the abstract.

Inconsistent capitalization of article titles and variation in the use of journal name abbreviations versus full titles are present in the references. These formatting styles should be standardized throughout the bibliography.

In the sentence: “We collected 4.86 terabases of HiFi reads from 49 8M SMRT cells sequenced on Sequel IIe and 41 25M SMRT cells sequenced on Revio”, how many cells does the author used?

Version 1:

Reviewer comments:

Reviewer #1

(Remarks to the Author)

The authors gave adequate and helpful responses to the comments. We believe that the modifications and additions make the manuscript easier to read and understand.

We have only a few additional comments:

References 15 to 17 mentioned on line 41 could be moved to the end of the sentence ending with 'contribute to phenotypic variation', which would make it clear that these references specifically support this point.

Line 80. It is not the mean read N50 nor the mean Phred score that demonstrate the expected negative correlation. The sentence should probably be rephrased.

We now understand that Figure 1a is not intended to highlight a linear correlation between the Illumina coverage and the HiFi coverage (which seemed to be suggested by the dotted grey line), but rather to emphasize that the coverage levels are comparable.

Line 88. A reference to the Materials and Methods section would help clarify more quickly the meaning of "alignments suitable for variant calling".

Paragraph starting line 204. This paragraph is dedicated to the comparison of the genotypes obtained with joint-calling and with force-calling using Sniffles. Since this terminology is not standard and is not used in the Sniffles documentation or in related papers, a brief mention of what the authors mean by 'joint-calling' and 'force-calling' would greatly facilitate the reading. This would allow the force-calling approach used in the paper to be clearly distinguished from the force-call option of CuteSV, which is employed in the Sniffles paper cited by the authors (ref. 62). A reference to the GitHub code, which clarifies the process, would also make it easier to understand.

Line 244. The 23.57% is understood as the per-individual heterozygosity of small variants in the limited subset of the 6 individuals with more than 30X HiFi coverage. If it is the case it would be easier to understand if this were specified.

Figure 4. To the best of our understanding, the text in the caption for b/d "Nominal p-value" is not appropriate and should probably be "Beta coefficient"

Line 363. The (non-specialist) reader has the feeling that the observation suggests that an increased missing rate leads to a smaller p-value which is counterintuitive. The relationship between the "unbalanced proportion of missing genotypes" and the p-value could be explained in more detail.

Line 520. To the best of our understanding, why cohort size should approximate effective population size is not straightforward. And, unless we overlooked it, it is not mentioned in the cited reference (ref 52).

We would like to emphasize the importance of the code provided by the authors, which helps to understand the analysis and enables its reproduction on these or other datasets.

Reviewer #2

(Remarks to the Author)

I'm happy with the explanation and changes made by the author. No extra comments from me.

Reviewer #3

(Remarks to the Author)

The author has carefully addressed my previous concerns, resulting in significant improvements to the manuscript. I think the manuscript is currently ready to be published.

Version 2:

Reviewer comments:

Reviewer #1

(Remarks to the Author)

Reviewers' comments:

Reviewer #1 (Remarks to the Author):

The manuscript by Mapel et al. describes an analysis, in cattle, that investigates the impact of structural variations on molecular phenotypes using a QTL approach. The work involved the production of long reads HiFi sequences for 120 bulls and the detection of small variants as well as structural variants from the alignment of those reads against the ARS-UCD2.0 genome assembly. Both small variants and structural variants (SV) are used to detect quantitative trait linkage with gene expression and gene splicing, an approach called molQTL. The main results of the paper include comparative analysis of short (Illumina) and long (HiFi) reads technology for small variant discovery, the added value of force-calling approach for genotyping structural variants in comparison to the genotypes provided by the Sniffles software, and, what is probably the main result of the paper, enrichment in structural variants of the top eQTL and sQTL variants. The analyses presented were conducted using appropriate methods, and the results shed light on the contribution of structural variation to molecular phenotypic variability in cattle. However, some aspects of the analyses performed and the way the results are presented raise several concerns.

We thank the reviewer for their comments.

1. The result section starts with a long description (almost 3 pages including figure 1) of a comparative analysis of short versus long read sequences for the purpose of detecting small variants. This question has been thoroughly addressed, using the same technologies (Illumina and HiFi) in the paper by Kosugi and Terao (ref 9 in the manuscript). The authors should explain the added value of their analysis and refer to the paper of Kosugi and Terao to explain the added value.

Given we had Illumina and HiFi sequencing on all 120 animals, we wanted to perform a large scale comparison on small variant calling. In particular, quantifying the relative variant calling accuracy in “normal”, tandem repeats, and transposable element regions. Comparisons between short and long reads for small variant calling are indeed becoming common. However, many papers are largely focused on well-studied human benchmarked samples, rather than large non-human cohorts. In the case of Kosugi and Terao 2024 [1], they examine two benchmarked human samples at 30x coverage, and examine short tandem repeats and segmental duplications as region annotations. They also conclude their paper with: “However, the conclusions of this study may be limited to human data because GATK requires known SNV/indel sites for VQSR/BQSR, and deep learning-based algorithms such as DeepVariant, PEPPER, and NanoCaller require custom models trained for nonhuman species.” As such, we have added additional text to the introduction, to highlight the added value of our results examining a larger cohort in a non-human species.

L45: “However, benchmarking results from human studies may not generalize to other species, since most of the widely applied variant genotyping algorithms are trained on human data and may perform suboptimally elsewhere [1–3].”

2. The scatter plot (Figure 1a) describing the relationship between short and long reads sequencing depth does not make sense in our point of view, coverage depth is a choice made when generating the data. The information provided by the plot is that some samples have an HiFi coverage significantly higher than the Illumina coverage. Difference in coverage between samples, and its impact on variant detection, is however not addressed in the analysis.

Although a few samples have higher coverage of HiFi or Illumina sequencing, the majority of samples have comparable coverage across the two technologies. A Wilcoxon signed-rank test of the coverages gives a p-value=0.077, suggesting there is not a statistically significant difference in sample coverage. However, as pointed out, the comparison of coverage on variant detection is not a focus of this work and so we have not added these details to the revised text and instead point out:

L78: "... although coverage differed substantially between both sequencing technologies for some individuals"

3. By addressing specifically "high impact variant" as predicted by Ensembl Variant Effect Predictor (EVEP), the authors imply that these variants may have specific characteristics that influence (short versus long read) variant detection. If that is indeed the case, the authors should explain why.

Without a standardized truth set to validate which callset (HiFi or Illumina) is more correct and the focus of this manuscript on QTL, we turned to variant annotations to further investigate whether the additional variants called by HiFi were meaningful. We have added some text to highlight this motivation

L124: "We hypothesized that the improved mappability of HiFi reads and the larger number of called variants provide access to previously unidentified genetic variants that may alter gene products or are deleterious to protein function."

There was not a statistically significant difference in the proportion of HIGH impact variants (HiFi: 9.8e-5; Illumina: 9.7e-5; Fisher's exact test p=0.731 on [[HiFi_HIGH,HiFi_SNP],[Illumina_HIGH,Illumina_SNP]]), suggesting the difference is not that one technology better captures certain characteristics, but that even the challenging regions containing the additional HiFi variants uncalled by Illumina can have functional consequences.

4. The authors mention an "observed improvements in small variant genotyping from long reads" that, to the best of our understanding, is not quantified nor really supported by the presented analysis (last sentence before addressing the detection of SV in the result section). An increased number of variants is observed in chromosome X, and 15 high impact variants not shared by both technologies seem to be associated with poor mapping of illumina reads but this does not imply an observed improvement in small

variant genotyping on the large set of detected variants (~20 millions).

Similar to the previous comment, without a truth set it is hard to completely quantify an improvement in variant calling accuracy. We have added the following text to acknowledge this limitation.

L143: "Previous evaluations suggest that, for small variants called from samples with moderate sequencing depth (8-15x), the misgenotyping rate may be as high as 2-5% [4-6]. For the present dataset, none of the 120 samples had microarray-called genotypes, thereby precluding a formal assessment of genotyping accuracy against a truth set. However, since previous research showed that DeepVariant calls cattle genotypes accurately from short reads [3], we considered the short-read variants as a truth and the HiFi variants as query"

We have also modified the original quoted text, highlighting that the long read alignments are improved over short read alignments, and that more small variants were called (especially in tandem repeat or otherwise repetitive regions). Again, given the manual inspection of callset discrepant variants in Supplementary Table 1 and literature referenced in the manuscript, we are confident (although unable to prove explicitly) these HiFi called variants are an improvement over Illumina called variants.

L158: "Given the observed improvements in long read alignments and more small variants called genome-wide, particularly in challenging regions, we only retained the HiFi-based small variants for downstream use."

We further investigated the allele frequency distribution of small variants called from HiFi and Illumina to clarify the characteristics of each set. The observed pattern suggests a higher error rate in the variants called from Illumina. We refer to a new Supplementary Figure 2c,d in the main text as follows:

L110: "A substantially higher proportion of singletons among the Illumina-only variants may indicate an elevated variant discovery error rate in short-read alignments (Supplementary Figure 2c,d)."

Supplementary Figure 2c) Variant calling with DeepVariant identified 21,015,811 biallelic autosomal variants in the HiFi and Illumina alignments of the 120 samples, of which 91.61% ($n=19,252,149$) were common to both sequencing technologies. The number of private variants was twice as high in the HiFi ($n=1,187,243$) than Illumina set ($n=576,419$). c) Of the 19.25 M common variants, 8.44% and 8.32% were detected as singletons with Illumina and HiFi, respectively (i.e., they were identified in the heterozygous state in only one individual). More than half (53.94%, $n=310,937$) of the Illumina-only variants were singletons, whereas the HiFi-only variants contained only 18.19% ($n=215,924$) singletons. This pattern suggests a higher error rate in the variants called from Illumina.

5. The second analysis presented in the results section deals with structural variant detection. Here, in contrast, no comparative analysis with SV detection using short reads is presented. We understand that this is not the focus of this work, but highlighting the added value of long reads for SV detection seems important in a study emphasizing the added value of considering SV detected by long reads to molQTL detection. The authors have in hand a unique dataset of 120 bulls sequenced with both short Illumina reads and long HiFi reads. Including a comparison of the variant catalogs from the two technologies on a dataset made on a large number of individuals would certainly greatly enhance the paper. The authors have however already addressed this question in reference 19. A correspondence between the two studies could be made by the authors in order to enable readers to fully benefit from the results presented in both studies. A better balance between the detection of small variants and the detection of

structural variants would be desirable.

As demonstrated in many recent papers [1, 7–10], structural variant detection with short reads is dominated by false positives and false negatives and particularly fails on insertion SVs. Given these limitations are so well established, we don't think a comparison of SVs called from Illumina vs. HiFi adds to the manuscript. However, we followed the reviewer's suggestion and better cross-reference studies that were conducted previously in the molQTL cohort:

L53: "Association testing with molecular phenotypes and SVs called from moderate-coverage short-read sequence data (average of 13.8-fold) has identified hundreds of SVs that impact gene expression and splicing in cattle testis tissue [8, 11]. However, short-read datasets are limited in their ability to accurately identify and genotype SVs, particularly large or multiallelic insertions and deletions [8–10]. Cohort-scale long-read sequencing in cattle has yet to be conducted to reveal the true impact of SVs on gene expression and splicing"

L73: "Short-read data from the same cohort were previously utilized to genotype small and structural variants for association analyses with molecular phenotypes [8, 11, 12]"

6. The authors implicitly suggest a relationship between the effective population size of the Braunvieh breed (70) and the fact that from the 100th sample onwards, approximately 100 new SVs are observed per additional sample (in Supplementary Figure 5 they even specify that this occurs starting from the 88th sample onwards). For a reader not familiar with population genetics, the connection between the two numbers is difficult to understand (70 and 100 or 70 and 88). If the relationship is simple it should be explained or a reference could be given. If the relationship is not straightforward it is probably better to omit the precise value of the effective population size to avoid disrupting the reading.

We agree that relating the effective population size to values like 100 novel SVs per sample is not straightforward and have removed the precise value.

L164 "This suggests that we likely captured the bulk of non-rare SVs present in the Braunvieh breed "

7. It is mentioned that 37,380 novel autosomal SVs were found compared to a previous pangenome-derived panel (constructed from 15 (?) Braunvieh haplotypes). Reference 19 should probably be mentioned here (in ref 19, a pangenome based on 16 assemblies is mentioned) The relationship between the two populations (ref 19 and this work) is not explained. We believe that some variants found in the pangenome-based approach were missed by the present one. This could be simply mentioned.

We have expanded this section to clarify the 16th animal is from the Piedmontese breed and not Braunvieh (see ref [11]), hence we excluded it. We also clarified that all 15

Braunvieh haplotypes used in the pangenome-panel are not related to samples we consider in this work. We have further added text to help explain that the SVs found only in the pangenome-panel are likely a combination of private SVs to those animals and differences in calling SVs with assembly-to-reference (pangenome-panel) or read-to-reference (this work) approaches.

L174: *“We found 37,380 novel autosomal SVs compared to a previous autosome-only, pangenome-derived SV panel constructed from one Piedmontese and 15 Braunvieh haplotypes not included in the HiFi-cohort analysed here, including 19,507 SVs with a minor allele frequency (MAF) between 1% and 15% (Figure 2b), as well as 376 duplications and 422 inversions which cannot be easily genotyped through the previously applied k-mer approach [11]. The 11,280 SVs present only in the pangenome-panel are likely private to those animals or related to differences in assembly-versus-alignment based SV calling.”*

8. The authors underline the benefits of force-calling to obtain reliable genotypes. It should probably more clearly stated in the text that force-calling is a feature offered by Sniffles.

We have made it explicit in the results section that we used sniffles2 to force-call the candidate SVs.

L206: *“Force-calling candidate SVs with sniffles2 from the original alignments...”*

9. The fact that “Force-calling SVs with sniffles2 overwhelmingly filled in sporadically missing genotypes” is, to the best of our understanding, not a surprise since the corresponding SVs were not found in the associated samples and could therefore not be genotyped (there is no corresponding line in the vcf file of those samples). If this is the case, this should be explained in the text by the authors.

Sniffles2 implements different thresholds in the de novo and force-calling modes, such that there may be insufficient read support to call the variant originally but sufficient read support to force-genotype that variant. In cases where there truly was no coverage, the SV call remains missing (the force-called SVs had a mean missingness of 1.2%). We have clarified this in the text

L217: *“The overwhelming majority of these changes (89.7%) were newly filled missing values (where there was potentially enough read support to force-genotype the SV but not call it originally) ...”*

We also clarify that the force-called genotypes may have slightly lower concordance.

L226: *“Force-called genotypes have been reported to be slightly less concordant than joint-called genotypes [13], although this negative consequence is very likely outweighed by the vast decrease in missingness.”*

While we were unable to directly assess the concordance of the raw and force-called SV genotypes due to the lack of a truth set, we compared per-sample heterozygosity between SV [raw and force-called] and SNP genotypes. Results agree with English et al. [13], and are shown in a newly compiled Supplementary Figure 10b,c. We refer to this figure in the main text:

L237: " The absence of a validated truth set of SV genotypes prevents a direct assessment of the SV genotyping accuracy achieved with sniffles2. Nonetheless, the autosomal per-individual heterozygosity was significantly higher for small variant (23.99%) than SV genotypes called from the HiFi reads regardless if the de-novo (22.57%, Welch's t-test p-value = 3.45×10^{-11}) or force-called (21.96%, Welch's t-test p-value = 4.99×10^{-19}) SV genotypes were used (Supplementary Figure 10b,c). The discrepancy between small variant- and SV-based heterozygosity is less evident in six samples with more than 30-fold HiFi coverage (de-novo: 23.57% vs. 22.69%, Welch's t-test p-value = 0.07; force-called: 23.57% vs. 22.52%, Welch's t-test p-value = 0.04), suggesting that high sequencing depth is required to comprehensively call heterozygous genotypes with sniffles2 and that heterozygous undercalling is more pronounced in force-called genotypes."

Supplementary Figure 10. Impact of force-calling SV genotypes with sniffles2. b-c) Scatterplots for the per-sample heterozygosity estimated from denovo SV calls and SNPs (b), and for the per-sample heterozygosity estimated from forced SV calls and SNPs (c). The lighter colours indicate the few samples with high coverage (>20 -fold). The fit of a linear model regressing SNP-based heterozygosity on the SV-based heterozygosity was slightly better for the denovo (adjusted R2: 0.91, p= $3.51e^{-64}$) than the forced SV (adjusted R2: 0.90, p= $2.12e^{-61}$) calls..

We also present a revised Figure 2c in the main manuscript, which shows genotype missingness on a per-site basis, and not per-individual as previously. The corresponding text in the main part has been updated.

L204: "Joint-calling SVs across the entire cohort resulted in a relatively high proportion of missing genotypes (mean of 5.3%) per-site, with duplications having the highest missingness (mean of 10.6%). Force-calling candidate SVs with sniffles2 from the original alignments substantially reduced the missingness to a mean of 1.3% (with the lowest mean missingness now observed in duplications at 0.9%), slightly lower than the missingness of joint-called SNPs (Figure 2c)."

Figure 2c) Forced genotyping of SVs substantially reduces per-site missingness to similar levels identified in SNP variant calling.

We also clarify, how the missingness filter before molQTL mapping impacts the variant set.

L261: "The applied filtering substantially altered the characteristics of the variant set used for molQTL mapping, reducing the average per-site missingness to 1.21 ± 1.80 for small variants and 0.03 ± 1.07 for SVs. Among the variants included in the association analyses, 43.1% of small variants and 88.5% of SVs exhibited complete genotype data across all 117 samples."

10. The precise meaning of unique variants, top variants and associated unique variants is not explained (Table 2). This should be stated explicitly to prevent the reader from having to infer it or search the literature.

We have added definitions for e/sGenes, e/sQTL, e/sVariants, and top variants to the Table 2 description.

L267: "e/sGenes (gene that contains at least one eQTL or sQTL) and e/sQTL (independent-acting signals) include both small variants and SV. e/sVariants are defined as a variant that passes the gene's significance threshold, with the total number of unique variants listed in parentheses. Top variants are defined as the most significant variant for each independent-acting eQTL or sQTL, with the number of unique variants listed in parentheses."

11. molQTL detection with small variants and SV in Braunvieh were already addressed in a previous study by the authors (ref 19). A correspondence between the two studies should be made by the authors in order to enable readers to fully benefit from the results presented in both studies.

We have expanded on the previous two studies in the Introduction to give more context to the work that has been done previously and how it differs to the current manuscript.

L53: *“Association testing with molecular phenotypes and SVs called from moderate-coverage short-read sequence data (average of 13.8-fold) has identified hundreds of SVs that impact gene expression and splicing in cattle testis tissue [8, 11]. However, short-read datasets are limited in their ability to accurately identify and genotype SVs, particularly large or multiallelic insertions and deletions [8–10]. Cohort-scale long-read sequencing in cattle has yet to be conducted to reveal the true impact of SVs on gene expression and splicing.”*

Also, we have added emphasis that short-read data from this cohort has already been collected to the Results and give an explanation for not directly comparing the current dataset to SVs identified with the pangenome-panel.

L73: *“Short-read data from the same cohort were previously utilized to genotype small and structural variants for association analyses with molecular phenotypes [8, 11, 12]”*

L179: *“The 11,280 SVs present only in the pangenome-panel are likely private to those animals or related to differences in assembly-versus-alignment based SV calling.”*

12. What do the authors mean by “feature’s significance threshold” ? (penultimate paragraph of the results section).

We have changed this instance to *“gene’s significance threshold”* (L402).

13. In Figure 4e, the black diamonds seem to exhibit a linear relationship which is not discussed in the text.

The linear relationship in Figure 4e is consistent with the expectation that heterozygotes have approximately 1.5-fold higher coverage than the chromosomal background. We have added additional context to the Figure 4 caption to explain the observed relationship. Notably, the samples misgenotyped as homozygous reference for this duplication have moderate coverage, indicating that coverage depth is not the primary driver of duplication genotyping errors.

L374: *“As expected, heterozygotes demonstrated on average an 1.51-fold (between 1.43 and 1.62-fold) coverage increase across the duplication relative to chromosome 2 background coverage.”*

14. The impact of genotyping - both structural variants (SVs) and small variants - on the ability to detect molQTLs is a very interesting point, and directing readers to relevant literature on this topic would be highly valuable.

We have added a sentence to the Introduction that emphasizes the importance of accurate genotyping for QTL detection and cite Abecasis et al. (2001), which investigates this with genotyped small variants. We have also added a sentence to the Discussion that highlights results from Chiang et al. (2018), which observed that SV genotype accuracy had a profound effect on SV eQTL detection.

L29: "Genome-wide association studies (GWAS) have identified numerous loci linked to specific traits but require accurate genotypes to detect putative causal variants and estimate allele substitution effects [14]."

L495: "Errors in SV genotyping can have a substantial impact on QTL detection; for example, Chiang et al. [15] reported that a 5% increase in error rate reduced SV eQTL detection by 19.6%. Almost one-third of the manually examined molQTL and all manually examined large duplications were affected by imperfectly genotyped small variants and SVs. The per-variant error rate among the manually examined molQTL was relatively low. However, because our sample size was small, even a few misgenotyped samples could strongly influence molQTL detection. We also observed instances, e.g., for large duplications, where much higher per-variant error rates obscured SV molQTL detection."

15. In the discussion, the authors indicate that they suspect long SVs (>15,000 bp) are missed, given the HiFi read lengths. The authors could mention that this applies to insertions (and duplications) and not to all SV (large deletions are detected as shown in Figure 2).

Yes, this does not apply to all SVs, only insertions and duplications. We now mention insertions and duplications specifically (L466).

16. Is the X chromosome really neglected in GWAS studies due to its repetitiveness?

We agree that there are more reasons for excluding the X chromosome in GWAS and expanded our point.

L465: "Long-read sequence data also improved alignments to the X and Y chromosomes—which are often neglected in GWAS for several reasons, including poor genotyping resulting from alignment errors due to their repetitiveness—and thereby facilitated the identification of over one hundred thousand small variants that were previously inaccessible with short-read sequencing, including 7 SNPs predicted as HIGH impact. However, variants along the MSY remained poorly characterized due to uneven coverage and copy-number complexity, limiting the ability to identify HIGH impact variants and molQTL for genes within this region."

17. The ENA ERP126174 project (last updated in 2021) contains 24 runs (some of which are Illumina reads) and is already mentioned in ref19 and does not contain the Hi-C reads analyzed in the article. The HiFi reads from the 120 bulls should be made publicly available.

All data are deposited in public repositories, and the accession IDs are listed in Supplementary Table 4.

See our Data Availability statement (L651): “DNA and RNA sequencing data of the analysed cohort are available in the ENA database at the study accessions PRJEB42335 (Long-read sequencing data from cattle for the purpose of de-novo genome assembly), PRJEB28191 (Short-read sequencing of cattle) and PRJEB46995 (Testis transcriptome of mature bulls). Accession identifiers for all samples are available as Supplementary Table 4”

Here are ENA screenshots documenting that the data are available online:

Search results for PRJEB42335

- Read
 - Experiment (24)
 - Run (24)

- Study
 - Study (1)
 - Project (1)

Experiment View all 24 results.	
ERX4849492	Sequel II sequencing
Run View all 24 results.	
ERR13636450	Illumina NovaSeq 6000 sequencing
Study	
ERP126174	Long-read sequencing data from cattle for the purpose of de-novo genome assembly
Project	
PRJEB42335	Long-read sequencing data from cattle for the purpose of de-novo genome assembly

Search results for PRJEB28191

- Read
 - Experiment (472)
 - Run (472)

- Study
 - Study (1)
 - Project (1)

Experiment View all 472 results.	
ERX6101198	Illumina NovaSeq 6000 paired end sequencing
Run View all 472 results.	
ERR2743212	Illumina HiSeq 2500 paired end sequencing
Study	
ERP110367	Short read sequencing of cattle
Project	
PRJEB28191	Short read sequencing of cattle

Search results for PRJEB46995

- Read
 - Experiment (511)
 - Run (511)

- Study
 - Study (1)
 - Project (1)

Experiment View all 511 results.	
ERX6136640	Illumina NovaSeq 6000 paired end sequencing
Run View all 511 results.	
ERR10186685	Illumina NovaSeq 6000 sequencing
Study	
ERP131230	Establishing an expression QTL cohort to detect molecular phenotypes in testis tissue of mature bulls
Project	
PRJEB46995	Establishing an expression QTL cohort to detect molecular phenotypes in testis tissue of mature bulls

The work presented here constitutes a contribution to the understanding of the impact of structural variation on gene expression and splicing. We believe that the paper would

greatly benefit from a rewrite of the Results section, placing less emphasis on raw numbers (such as the number of SVs, percentages, and p-values) and instead focusing more on the questions addressed and how the data help to answer them.

AU: Thank you for appreciating the relevance of our work. We made an effort to provide more context at several places in the results and discussion sections to ensure the manuscript is well accessible.

Reviewer #2 (Remarks to the Author):

This manuscript by Mapel et al. presents a comprehensive analysis of structural variants (SVs) and their impact on molecular phenotypes in cattle using long-read HiFi sequencing. The authors sequenced 120 *Bos taurus taurus* bulls of primarily Braunvieh ancestry using PacBio HiFi technology, generating 4.86 terabases of data with an average read N50 of 16.3 kb at 13.5-fold coverage. This represents one of the largest long-read datasets available for a non-model organism and addresses a critical gap in cattle genomics where previous SV studies have relied primarily on short-read sequencing, pangenome analyses, or QTL fine-mapping approaches.

The study's strength lies in its integrative approach, combining comprehensive variant discovery with functional characterization through molecular QTL (moQTL) mapping using deeply sequenced total RNA from testis tissue of the same individuals. This design enables direct assessment of SV impacts on gene expression and splicing variation, moving beyond mere variant cataloging to functional interpretation.

Overall Assessment

The manuscript is very well-written and logically organized, with clear figures and comprehensive supplementary materials. The methods section provides sufficient detail for reproduction, and the results are presented systematically. However, I have several major points for the author team to address/clarify before accepting the MS.

We thank the reviewer for their encouraging comments. We have expanded on some results, particularly those regarding missing genotypes and misgenotyping, to add additional context and clarity.

Major Comments

- Force-calling substantially improved missing data rates, but you report that 4% of genotypes changed during this process. How did you validate that these changes represent improvements rather than the introduction of new errors?

The vast majority of changed genotypes (nearly 90%) are filling missing genotypes, while the remaining ~10% are changing existing calls. Many of the cases of changing existing calls occur in complex regions, where even manually assigning genotypes to complex, clustered SVs is challenge (as shown already in Sup Fig 11).

As discussed in a comment from Reviewer #1, force-genotyping may have lower thresholds for e.g., read support to assign a genotype compared to de novo calling. The per-individual heterozygosity (estimated from all autosomal variants with MAF>0) is slightly higher correlated between SNPs and the denovo SV calls than SNPs and forced SV calls. These results agree with English et al. [13].

L226: *“Force-called genotypes have been reported to be slightly less concordant than joint-called genotypes [13], although this negative consequence is outweighed by the vast decrease in missingness.”*

These results are shown in a newly compiled Supplementary Figure 10b,c. We refer to this figure in the main text:

L237: *“The absence of a validated truth set of SV genotypes prevents a direct assessment of the SV genotyping accuracy achieved with sniffles2. Nonetheless, the autosomal per-individual heterozygosity was significantly higher for small variant (23.99%) than SV genotypes called from the HiFi reads regardless if the de-novo (22.57%, Welch's t-test p-value = 3.45×10^{-11}) or force-called (21.96%, Welch's t-test p-value = 4.99×10^{-19}) SV genotypes were used (Supplementary Figure 10b,c). The discrepancy between small variant- and SV-based heterozygosity is less evident in six samples with more than 30-fold HiFi coverage (de-novo: 23.57% vs. 22.69%, Welch's t-test p-value = 0.07; force-called: 23.57% vs. 22.52%, Welch's t-test p-value = 0.04), suggesting that high sequencing depth is required to comprehensively call heterozygous genotypes with sniffles2 and that heterozygous undercalling is more pronounced in force-called genotypes.”*

Supplementary Figure 10. Impact of force-calling SV genotypes with sniffles2. b-c) Scatterplots for the per-sample heterozygosity estimated from denovo SV calls and SNPs (b), and for the per-sample heterozygosity estimated from forced SV calls and SNPs (c). The lighter colours indicate the few samples with high coverage (>20 -fold). The fit of a linear model regressing SNP-based heterozygosity on the SV-based

heterozygosity was slightly better for the denovo (adjusted R^2 : 0.91, p = 3.51e-64) than the forced SV (adjusted R^2 : 0.90, p = 2.12e-61) calls.

- The reported 27–32% misgenotyping rate is derived from the top 1% largest-effect molQTL, I have a feeling this is only valid as a “worst-case” indicator in the most error-prone, high-effect loci, but it likely overestimates the genome-wide rate.

A systematic validation of genotype calls is only possible with a standardized truth set such as human GIAB. Such a truth set is not available in cattle, and so assessing genotype accuracy at scale is not possible. It is important to keep in mind, that our definition of «misgenotyping» refers to variants with at least one erroneous genotype in the 117 samples. We clarify this in the revised manuscript.

L406: "Thus, while nearly a third of these variants were affected by genotyping errors, the extent of misgenotyping per variant was low; for most of the manually examined SVs, only one or two samples showed genotyping errors, corresponding to a per-variant error rate of approximately 2%."

L499: "The per-variant error rate among the manually examined molQTL was relatively low. However, because our sample size was small, even a few misgenotyped samples could strongly influence molQTL detection. We also observed instances, e.g., for large duplications, where much higher per-variant error rates obscured SV molQTL detection."

Earlier studies have shown that the concordance between sequence-called and microarray-derived SNPs is in the order of 97-98% for samples with 13-fold coverage [4–6]. In a previously analysed cohort of 49 samples that had both sequence-called and microarray-derived small variants [5], nearly 35% of the SNPs had at least one erroneously genotyped site in at least one sample. However, the number of SNPs with genotyping errors in more than one sample is low (12%), and less than 1% of all sites have genotyping errors in more than 10% of the samples. Sequencing read data of these animals are available from the European Nucleotide Archive (ENA) (<http://www.ebi.ac.uk/ena>) under primary accession PRJEB28191

Concordance between microarray-derived and DeepVariant-called sequence-variant genotypes on bovine chromosome 26 in a previously analysed cohort of cattle.

Proportion of SNPs with at least one genotype error in 49 analysed samples.

These investigations indicate that the reported 27-32% misgenotyping rate is a reasonable estimate given the ~13.5-fold sequencing coverage, and unlikely to represent a worst-case scenario.

We refer to this work in L143: "*Previous evaluations suggest that, for small variants called from samples with moderate sequencing depth (8-15x), the misgenotyping rate may be as high as 2-5% [4-6].*"

- Could the authors comment on their approach to assign heterozygous and homozygous genotypes for duplications? Fixed fold-change cutoffs might have their own caveats. In addition, could the author provide their thoughts on duplications and how well we can use them to predict impact ?

The genotypes were assigned by sniffles2. For large duplications overlapping genes, we used a coverage cutoff estimated from short read sequencing based on the expectation that heterozygotes and homozygotes have approximately 1.5-fold and 2-fold higher coverage than the chromosomal background, respectively. We agree that this approach has caveats but should still work reasonably well for large duplications. This is now clarified in the revised manuscript.

L604: "*Heterozygous and homozygous genotypes were assigned to the sniffles2-called duplications when the average coverage over the duplicated sequence was at least 1.3-fold and 1.8-fold higher, respectively, than the average coverage outside the duplication on the chromosome containing the duplication*"

L374: "*As expected, heterozygotes demonstrated on average an 1.51-fold (between 1.43 and 1.62-fold) coverage increase across the duplication relative to chromosome 2 background coverage.*"

- Is it possible re-estimate SV vs SNP effect sizes using a high-confidence SV subset (e.g., non-TR loci, size < 10 kb, strong read support and breakpoint consistency). This will help determine whether the observed effect-size equivalence is driven by genotyping error.

To address this, we restricted the SV-molQTL set to variants < 10 kb, not annotated as tandem repeats (TR), with length deviation < 10 and positional deviation < 5. Under these criteria, SVs showed slightly larger eQTL effect sizes than small variants (Wilcoxon p =

0.00361), whereas small variants had larger sQTL effect sizes than SVs (Wilcoxon $p = 0.00022$).

Although these findings suggest that genotyping errors may influence molQTL effect-size estimates, they do not establish that such errors drive the differences. We observed genotyping errors across a wide range of SV sizes and classes, complicating the construction of a uniformly high-confidence SV subset. Cohort-scale, higher-coverage long-read data—or advances in genotype imputation—will be needed to resolve the true effect-size differences between small variants and SVs.

- Based on your experience with genotyping errors and coverage limitations, what specific recommendations would you make for future long-read SV studies in livestock?

Our analyses show that SV genotyping benefits from higher sequencing coverage than the 13.5-fold achieved in our study. Although long-read sequencing has become much cheaper recently, a compromise between sample size and coverage needs to be made. Together with sophisticated phasing and imputation algorithms, these data will continue allowing fundamental discoveries. A recent study from The All of Us Research Program comes to a similar conclusion [16]. This has been added to the discussion.

L519: " *In the meantime, moderate-coverage long-read sequencing data, when collected in cohorts approximating the effective population size as demonstrated here, will enable the development of population-scale analytical methods such as SV genotype imputation [16].*"

- Given the focus on male fertility traits, what is the practical significance of your findings for cattle breeding? Can you quantify the potential economic impact of incorporating SV information into genomic selection programs?

Our manuscript investigates how much SVs contribute to the variability of complex traits. To address this question, SV genotype and phenotype data from a long-read sequenced cohort are needed. We used data from a cohort of 120 bulls that had long read sequencing data and molecular phenotypes from testis RNA sequencing. We examined the impact of testis e/sQTL on male fertility in previous work but don't investigate male fertility traits in this manuscript. Therefore, we can't connect our work to male fertility or practical implications of SVs. Regardless, our findings hint towards a relevant contribution of SVs to complex trait variation. We indicate this in the manuscript, while being careful to also mention our findings are based off a very specific tissue.

L476: "Despite being derived from a specific phenotype (testis RNA sequencing data), our findings support the broader importance of SVs in complex trait variation, as previously reported in cattle and other species."

- Figure quality: Some supplementary figures have poor legends and inconsistent formatting (detailed in minor points below).

Fixed.

The absence of line numbers makes it very difficult to specify where corrections are needed. Below are some minor points.

Our apologies for not including line numbers in the initial submitted draft. We included them in our revision and refer to them when necessary.

Minor Points

- Consider creating an abbreviation table for all gene names (e.g., STK39, MAGI3, CERS6).

We have added a table (Supplementary Table 5) that lists the NCBI RefSeq abbreviation for each gene that appears in the main text, including its full description and biotype.

- Scan the manuscript again for legend abbreviations/annotations, several time I can find figure has abbreviations but there was no annotation in the legends (e.g., VNTR, MSY).

Fixed.

- "We identified 23 SV eQTL that were not in strong LD ($r^2 \geq 0.8$)" – For some reason I found several r^2 superscripts are hyperlinked to citations.

Fixed.

- Typo: "ploy-G-tails" → "poly-G-tails"

Fixed.

- Please maintain consistent mathematical formatting:
"We considered genes with TPM ≥ 0.1 and at least 6 supporting reads in $\geq 10\%$ of samples for eQTL mapping."

Fixed.

- Keep consistent with "long-read" (with or without hyphen).

We have applied hyphenation for 'long read' and 'short read' consistently, using it only in adjectival forms and omitting it in noun forms, in accordance with Nature's formatting guidelines.

- Some references are inconsistently formatted:
Nat Commun vs. Nature Communications

Fixed.

- Abstract: "Sequencing mapping cohorts" → "Sequencing cohorts"

Fixed.

- Add a short cautionary note regarding interpretability of MSY molQTL given uneven coverage and copy-number complexity.

We have added a cautionary note to the Discussion.

L470: *"However, variants along the MSY remained poorly characterized due to uneven coverage and copy-number complexity, limiting the ability to identify HIGH impact variants and molQTL for genes within this region."*

- Introduction: "orders of magnitudes longer" → "orders of magnitude longer"

Fixed.

- Results: "on the order of 100s" → "on the order of hundreds"

Fixed.

- Supplementary Figure 1: Reports "Pearson's $r^2=0.65$, $p=5.9e-37$ " while main paper states "Pearson's $r: -0.80$, $p=5.87 \times 10^{-37}$ " – which is correct?

The difference was due to rounding between r and r^2 . We have consistently used r throughout the manuscript when referring to statistical correlations.

Reviewer #3 (Remarks to the Author):

Mapel et al. present a comprehensive study leveraging HiFi long-read sequencing in 120 Braunvieh bulls to generate an exhaustive catalog of both small variants (23.8 M) and structural variants (SVs; 79.3 k) in cattle testes. Combining these genomic data with total RNA-seq from testis tissue of 117 samples, the authors performed cis-molecular QTL (molQTL) mapping for expression (eQTL) and splicing (sQTL). The present study offered a huge amount of data and candidate loci contributed to complex traits. However, there are still some questions required to be addressed:

We thank the reviewer for their comments.

All molQTL analyses are derived exclusively from testis RNA. Regulatory landscapes differ markedly across tissues (e.g. liver, muscle, mammary gland), so conclusions about SV impacts on "complex traits" may not extrapolate to traits governed by other tissues. Additional RNA-seq from diverse tissues would strengthen claims of SV enrichment genome-wide.

Indeed, our association tests only considered molecular phenotypes from testis tissue. The purpose of our study is to investigate the influence of variants identified through long-read sequencing on direct phenotypes (gene expression and splicing estimates) from the same individuals. These samples were collected previously (as described in Mapel et al. [12]) and biobanked, thus we only had access to testis tissue. We have added additional emphasis that we only consider testis tissue to the Abstract (L19) and the Discussion. Moreover, studies that investigated the influence of SVs on gene expression with short-read whole-genome sequence data have shown that the proportion of molQTL with SV lead variants is consistent across tissue types (ranging from 2.4 to 4.5%; [15]). Enrichment of SVs in molQTL has also been described in other tissue types and species [17–19]. Thus, we suspect that the enrichment of SVs as lead variants detected in testis likely mirrors that in other tissues.

L476: "Despite being derived from a specific phenotype (testis RNA sequencing data), our findings support the broader importance of SVs in complex trait variation, as previously reported in cattle and other species."

SVs prioritized as putative causal variants (e.g. ATXN7L3B deletion, ADGRF1 splice-junction deletion) are identified solely through statistical association and manual inspection. Functional validation (e.g. reporter assays, CRISPR perturbation) is needed

to confirm their regulatory effects. Without such validation, mechanistic claims remain speculative. If this is impossible at this stage, maybe discuss it in the discussion part.

As further functional experiments are not feasible at this stage, we have added a sentence about functional validation to the Discussion.

L486: *“Nonetheless, confirming a causal role for these SVs requires further functional experiments.”*

What are small variants in the abstract, SNP? InDels or other else?

Yes, these are SNPs and small INDELS. We have added this to the Abstract (L14).

What does the author mean about “the top variants” in the abstract.

“Top variants” are the most significant variant for an independent-acting molQTL. We changed to *“most significant”* in the Abstract (L17).

Inconsistent capitalization of article titles and variation in the use of journal name abbreviations versus full titles are present in the references. These formatting styles should be standardized throughout the bibliography.

We thank the review for drawing attention to errors in the bibliography. We are aware of some inconsistencies in the bibliography of the original and current version of the manuscript but will ensure they are rectified before typesetting.

In the sentence: “We collected 4.86 terabases of HiFi reads from 49 8M SMRT cells sequenced on Sequel IIe and 41 25M SMRT cells sequenced on Revio”, how many cells does the author used?

We used 49 SMRT cells (8M) from Sequel IIe and 41 SMRT cells (25M) from Revio. We have rephrased this sentence for clarity.

L76: *“We collected 4.86 terabases of HiFi reads sequenced from 49 and 41 Sequel IIe 8M and Revio 25M SMRT cells, respectively.”*

References in the rebuttal letter

1. Kosugi S, Terao C. Comparative evaluation of SNVs, indels, and structural variations detected with short- and long-read sequencing data. Hum Genome Var. 2024;11:18. <https://doi.org/10.1038/s41439-024-00276-x>.

2. Kalleberg J, Rissman J, Schnabel RD. Overcoming limitations to customize DeepVariant for domesticated animals with TrioTrain. *Genome Res.* 2025;35:1859–74. <https://doi.org/10.1101/gr.279542.124>.
3. Lloret-Villas A, Bhati M, Kadri NK, Fries R, Pausch H. Investigating the impact of reference assembly choice on genomic analyses in a cattle breed. *BMC Genomics.* 2021;22:363. <https://doi.org/10.1186/s12864-021-07554-w>.
4. Daetwyler HD, Capitan A, Pausch H, Stothard P, van Binsbergen R, Brøndum RF, et al. Whole-genome sequencing of 234 bulls facilitates mapping of monogenic and complex traits in cattle. *Nat Genet.* 2014;46:858–65. <https://doi.org/10.1038/ng.3034>.
5. Crysanto D, Wurmser C, Pausch H. Accurate sequence variant genotyping in cattle using variation-aware genome graphs. *Genetics Selection Evolution.* 2019;51:21. <https://doi.org/10.1186/s12711-019-0462-x>.
6. Jansen S, Aigner B, Pausch H, Wysocki M, Eck S, Benet-Pagès A, et al. Assessment of the genomic variation in a cattle population by re-sequencing of key animals at low to medium coverage. *BMC Genomics.* 2013;14:446. <https://doi.org/10.1186/1471-2164-14-446>.
7. Cameron DL, Di Stefano L, Papenfuss AT. Comprehensive evaluation and characterisation of short read general-purpose structural variant calling software. *Nat Commun.* 2019;10:3240. <https://doi.org/10.1038/s41467-019-11146-4>.
8. Bhati M, Mapel XM, Lloret-Villas A, Pausch H. Structural variants and short tandem repeats impact gene expression and splicing in bovine testis tissue. *Genetics.* 2023;225:iyad161. <https://doi.org/10.1093/genetics/iyad161>.
9. Lee Y-L, Bosse M, Takeda H, Moreira GCM, Karim L, Druet T, et al. High-resolution structural variants catalogue in a large-scale whole genome sequenced bovine family cohort data. *BMC Genomics.* 2023;24:225. <https://doi.org/10.1186/s12864-023-09259-8>.
10. Grant JR, Herman EK, Barlow LD, Miglior F, Schenkel FS, Baes CF, et al. A large structural variant collection in Holstein cattle and associated database for variant discovery, characterization, and application. *BMC Genomics.* 2024;25:903. <https://doi.org/10.1186/s12864-024-10812-2>.
11. Leonard AS, Mapel XM, Pausch H. Pangenome genotyped structural variation improves molecular phenotype mapping in cattle. *Genome Res.* 2024;:gr.278267.123. <https://doi.org/10.1101/gr.278267.123>.
12. Mapel XM, Kadri NK, Leonard AS, He Q, Lloret-Villas A, Bhati M, et al. Molecular quantitative trait loci in reproductive tissues impact male fertility in cattle. *Nat Commun.* 2024;15:674. <https://doi.org/10.1038/s41467-024-44935-7>.

13. English AC, Cunial F, Metcalf GA, Gibbs RA, Sedlazeck FJ. K-mer analysis of long-read alignment pileups for structural variant genotyping. *Nat Commun.* 2025;16:3218. <https://doi.org/10.1038/s41467-025-58577-w>.
14. Abecasis GR, Cherny SS, Cardon LR. The impact of genotyping error on family-based analysis of quantitative traits. *Eur J Hum Genet.* 2001;9:130–4. <https://doi.org/10.1038/sj.ejhg.5200594>.
15. Chiang C, Scott AJ, Davis JR, Tsang EK, Li X, Kim Y, et al. The impact of structural variation on human gene expression. *Nat Genet.* 2017;49:692–9. <https://doi.org/10.1038/ng.3834>.
16. Garimella KV, Li Q, Wertz J, Lee SK, Cunial F, Huang Y, et al. Population-scale Long-read Sequencing in the All of Us Research Program. 2025;:2025.10.02.25336942. <https://doi.org/10.1101/2025.10.02.25336942>.
17. Ho SS, Urban AE, Mills RE. Structural variation in the sequencing era. *Nat Rev Genet.* 2020;21:171–89. <https://doi.org/10.1038/s41576-019-0180-9>.
18. Vialle RA, de Paiva Lopes K, Bennett DA, Crary JF, Raj T. Integrating whole-genome sequencing with multi-omic data reveals the impact of structural variants on gene regulation in the human brain. *Nat Neurosci.* 2022;25:504–14. <https://doi.org/10.1038/s41593-022-01031-7>.
19. Jakubosky D, D’Antonio M, Bonder MJ, Smail C, Donovan MKR, Young Greenwald WW, et al. Properties of structural variants and short tandem repeats associated with gene expression and complex traits. *Nat Commun.* 2020;11:2927. <https://doi.org/10.1038/s41467-020-16482-4>.

Reviewers' comments:

Reviewer #1 (Remarks to the Author):

The authors gave adequate and helpful responses to the comments. We believe that the modifications and additions make the manuscript easier to read and understand. We have only a few additional comments:

AU: Thank you!

References 15 to 17 mentioned on line 41 could be moved to the end of the sentence ending with 'contribute to phenotypic variation', which would make it clear that these references specifically support this point.

AU: Done (line 41)

Line 80. It is not the mean read N50 nor the mean Phred score that demonstrate the expected negative correlation. The sentence should probably be rephrased.

AU: We have rephrased this section to indicate the correlation is between read N50 and Phred quality, rather than the means of those quantities.

L79: Read N50 (mean: 16.3 Kb) and Phred quality score (mean: 33.7) were strongly negatively correlated (Pearson's r : -0.80, $p=5.87 \times 10^{-37}$; Supplementary Figure 1), as expected for HiFi reads.

We now understand that Figure 1a is not intended to highlight a linear correlation between the Illumina coverage and the HiFi coverage (which seemed to be suggested by the dotted grey line), but rather to emphasize that the coverage levels are comparable.

AU: ok

Line 88. A reference to the Materials and Methods section would help clarify more quickly the meaning of "alignments suitable for variant calling".

AU: We have clarified the quality and coverage thresholds to "suitable" alignments more explicit.

L88: ...autosomal sequence covered by alignments suitable for variant calling (MAPQ>5 and read depth>2) for HiFi...

Paragraph starting line 204. This paragraph is dedicated to the comparison of the genotypes obtained with joint-calling and with force-calling using Sniffles. Since this terminology is not standard and is not used in the Sniffles documentation or in related papers, a brief mention of what the authors mean by 'joint-calling' and 'force-calling' would greatly facilitate the reading. This would allow the force-calling approach used in the paper to be clearly distinguished from the force-call option of CuteSV, which is employed in the Sniffles paper cited by the authors (ref. 62). A

reference to the GitHub code, which clarifies the process, would also make it easier to understand.

AU: We feel that these are relatively standard terms for population-level variant calling. For example in Sniffles, see joint-calling used here <https://github.com/fritzsedlazeck/Sniffles/wiki/SV-calling-with-Sniffles#population-sv-calling> and force-calling here <https://github.com/fritzsedlazeck/Sniffles?tab=readme-ov-file#d-genotyping-a-known-set-of-svs-force-calling>. In addition, there are further cases of joint or force calling/genotyping in many commonly used variant calling tools, like GATK (<https://gatk.broadinstitute.org/hc/en-us/articles/360035890431-The-logic-of-joint-calling-for-germline-short-variants>) or DRAGEN (https://support-docs.illumina.com/SW/DRAGEN_v310/Content/SW/DRAGEN/ForceGenotyping.html). As far as we can determine, CutSV force-call mode is the same concept of genotyping a specific set of pre-existing SV calls from alignments, as described here <https://github.com/tjiangHIT/cuteSV?tab=readme-ov-file#changelog> (which is now part of cuteFC rather than cuteSV). The reference to cuteSV in the Sniffles paper (ref 62), is indeed referring to genotyping a set of known SVs from earlier merging steps, which is the same process we followed in this manuscript (joint-call -> merge+filter -> force-call).

Line 244. The 23.57% is understood as the per-individual heterozygosity of small variants in the limited subset of the 6 individuals with more than 30X HiFi coverage. If it is the case it would be easier to understand if this were specified.

AU: Done, we now write:

L243: The discrepancy between the per-individual small variant- and SV-based heterozygosity is...

Figure 4. To the best of our understanding, the text in the caption for b/d “Nominal p-value” is not appropriate and should probably be “Beta coefficient”

AU: Thank you for noticing this. We have changed “Nominal p-value” to “Beta coefficient”.

Line 363. The (non-specialist) reader has the feeling that the observation suggests that an increased missing rate leads to a smaller p-value which is counterintuitive. The relationship between the “unbalanced proportion of missing genotypes” and the p-value could be explained in more detail.

AU: Done, we now write:

L362: These differences in p-values across multiple pairs of variants in perfect LD were often driven by uneven distributions of missing genotypes. Because each variant within a pair of variants in LD exhibited its own pattern of missingness, the linear regression models were effectively fitted on different subsets of individuals. This slight mismatch in sample composition led to minor differences in estimated effect sizes and standard errors, and consequently in the resulting p-values.

Line 520. To the best of our understanding, why cohort size should approximate effective population size is not straightforward. And, unless we overlooked it, it is not mentioned in the cited reference (ref 52).

AU: The reviewer is correct. The haplotype diversity covered by the cohort is much more important than the size. We now write:

L525: In the meantime, moderate-coverage long-read sequencing data, when collected from cohorts that adequately reflect the haplotype diversity of the target population as demonstrated here,

We would like to emphasize the importance of the code provided by the authors, which helps to understand the analysis and enables its reproduction on these or other datasets.

AU: We agree that sharing both data and code is important to facilitate reproducing results under FAIR principles. The computational workflows we used to analyse the data available through https://github.com/AnimalGenomicsETH/HiFi_cohort, which is mentioned in the Code availability section.

Reviewer #2 (Remarks to the Author):

I'm happy with the explanation and changes made by the author. No extra comments from me.

AU: Thank you!

Reviewer #3 (Remarks to the Author):

The author has carefully addressed my previous concerns, resulting in significant improvements to the manuscript. I think the manuscript is currently ready to be published.

AU: Thank you!

REVIEWERS' COMMENTS:

Reviewer #1: Everything is fine with me now.

Author: Thank you!